



# Modelling the Mara River Basin with data uncertainty using water levels for calibration

Petra Hulsman[1], Thom A. Bogaard[1], Hubert H.G. Savenije[1]

[1]Water Resources Section, Faculty of Civil Engineering and Geosciences, Delft University of Technology, Stevinweg 1, 2628 CN Delft, the Netherlands

Correspondence to: Petra Hulsman (p.hulsman@tudelft.nl)

**Abstract.** Hydrological models play an important role in Water Resources Management. In hydrological modelling, discharge data is generally required for calibration. To obtain continuous time series, water levels are usually converted into discharge by using a rating curve. However with this methodology, uncertainties are introduced in the discharge data due to insufficient observations, inadequate rating curve fitting procedures, extrapolation or temporal changes in the river geometry. Unfortunately, this is often the case in many African river basins. In this study, a semi-distributed rainfall runoff model has been applied to the Mara River Basin for the assessment of the water availability. To reduce the effect of discharge uncertainties in this model, water levels instead of discharge time series were used for calibration. In this model, seven sub-catchments are distinguished and four hydrological response units: forest, shrubs, cropland and grassland. To calibrate the model on water level data, modelled discharges have been converted into water levels using cross-section observations and the Strickler formula. In addition, new geometric rating curves have been obtained based on modelled discharge, observed water level and the Strickler formula. This procedure resulted in good and consistent model results during calibration and validation. The hydrological model was able to reproduce the water depths for the entire basin as well as for the Nyangores sub-catchment in the north. The geometric and recorded (i.e. existing) rating curves were significantly different at Mines, the catchment outlet, probably due to uncertainties in the recorded discharge time series. At Nyangores however, the geometric and recorded discharge were almost identical. In addition, it has been found that the precipitation estimation methodology influenced the model results significantly. Application of a single station for each sub-catchment resulted in flashier responses whereas Thiessen averaged precipitation resulted in more dampened responses. In conclusion, by using water level time series for calibrating the hydrological model of the Mara River Basin promising model results were obtained. For this river basin, the main limitation for obtaining an accurate hydrograph representation was the inadequate knowledge on the spatial distribution of the precipitation.



## 1 Introduction to rating curve uncertainties


Hydrological models play an important role in Water Resources Management. In hydrological modelling, discharge time series are of crucial importance. For example, discharge is used when estimating flood peaks (Di Baldassarre et al., 2012;Kuczera, 1996), calibrating models (Domeneghetti et al., 2012;McMillan et al., 2010) or determining the model structure (McMillan and Westerberg, 2015;Bulygina and Gupta, 2011). Discharge is

commonly measured indirectly through interpolation of velocity measurements over the cross-section (WMO, 2008;Di Baldassarre and Montanari, 2009). However, to obtain frequent or continues discharge data, this method is time consuming and cost-inefficient. In African river catchments, the quantity and quality of the available discharge measurement is unfortunately often inadequate for reliable hydrological modelling (Shahin, 2002;Hrachowitz et al., 2013).


There are several sources of uncertainty in discharge data when using rating curves that cannot be neglected. First, measurement errors in the individual discharge measurements affect the estimated continuous discharge data, for example in the velocity-area method uncertainties in the cross-section and velocity can arise due to poor sampling (Pelletier, 1988;Sikorska et al., 2013). Second, these measurements are usually done during

normal flows, however during floods the rating curve needs to be extrapolated. Therefore, the uncertainty increases for discharges under extreme conditions (Di Baldassarre and Claps, 2011;Domeneghetti et al., 2012). Thirdly, the fitting procedure does not always account well for irregularities in the profile, particularly when banks are overtopped. Finally, the river is a dynamic, non-stationary system which influences the rating curve: such as changes in the cross-section due to sedimentation or erosion, backwater effects or hysteresis (Petersen-

Øverleir, 2006). The lack of incorporating such temporal changes in the rating curve increases the uncertainty in discharge data (Guerrero et al., 2012;Jalbert et al., 2011;Morlot et al., 2014). As a result, the rating curve should be regularly updated to take such changes into account. The timing of adjusting the rating curve relative to the changes in the river affects the number of rating curves and the uncertainty (Tomkins, 2014).

The goal of this study is to develop a reliable hydrological model for the semi-arid and poorly gauged Mara River Basin in Kenya. Previous studies have focused on assessing the uncertainty of rating curves (Di Baldassarre and Montanari, 2009;Clarke, 1999) and their effect on model predictions (Karamuz et al., 2016;Sellami et al., 2013;Thyer et al., 2011). In this study however, the effects of discharge uncertainties are avoided by using water level instead of discharge time series for model calibration by incorporating the

hydraulic equation describing the rating curve within the model. The model results are verified using a few high quality discharge measurements. In previous studies, water level time series are found to provide valuable information on the flow dynamics for model calibration, especially in wet catchments whereas in dry catchments additional information is needed to constrain the flow volume (Seibert and Vis, 2016;Jiang et al., 2017).




## 2 Site description of the Mara River Basin and data availability

The Mara River originates in Kenya in the Mau Escarpment and flows through the Masai Mara National Reserve in Kenya into Lake Victoria in Tanzania. The main tributaries are the Nyangores and Amala Rivers in the upper reach and the Lemek, Talak and Sand in the middle reach (Fig. 1). The first two tributaries are

perennial while the remaining tributaries are ephemeral, which generally dry out during dry periods. In total, the river is 395 km long (Dessu et al., 2014) and its catchment covers an area of about 11,500 km$^2$ (McClain et al., 2013) of which 65% is located in Kenya (Mati et al., 2008).

Within the Mara River Basin, there are two wet seasons linked to the annual oscillations of the ITCZ (Inter-

tropical Convergence Zone). The first wet season is from March to May and the second from October to December (McClain et al., 2013). The precipitation varies spatially over the catchment following the local topography. The largest annual rainfall can be found in the upstream area of the catchment: between 1000 and 1750 mm/yr. In the middle and downstream areas, the annual rainfall is between 900 and 1000 mm/yr and between 300 and 850 mm/yr, respectively (Dessu et al., 2014).


The elevation of the river basin varies between 3000 m above sea level at the Mau Escarpment, 1480 m at the border to Tanzania and 1130 m at Lake Victoria (McClain et al., 2013). In the Mara River Basin, the main land cover types are agriculture, grass, shrubs and forests. The main forest in the catchment is the Mau Forest, which is located in the north. Croplands are mainly found in the north and in the south, whereas the middle part is

dominated by grasslands.

In the Mara River Basin, long term daily water level and discharge time series are available for 44-60 years between 1955 and 2015 at the downstream station near Mines and in two tributaries: the Nyangores and Amala. In addition, precipitation and temperature is measured at 29 and 5 stations, respectively (Fig. 1 and Table 1).

However, the temporal coverage of these data is poor as there are many gaps.

Also, there are many uncertainties in the discharge and precipitation data. Discharge data analyses indicated that the time series were unreliable due to various inconsistencies in the data, for example changing rating curves at Amala, unrealistic rating curve compared to cross-section based estimations at Nyangores and high scatter in the

discharge-water level graph at Mines; also back-calculated cross-section average flow velocities were much lower than field measurements at Mines. The precipitation data analysis showed a high spatial variability between the rainfall stations. This could be a result of high heterogeneity which is poorly represented by the limited number of stations available. See supplement for more details.

As a result of using this precipitation data for hydrological modelling, significant errors and uncertainties will occur in the modelled discharge which are required for a solid water resources allocation plan. The uncertainties in the measured rating curve and precipitation need to be taken into account in the evaluation of the hydrological model performance. In contrast to previous studies where discharge time series were used to calibrate the hydrological model of the Mara River Basin using the Soil Water Assessment Tool (SWAT) (Dessu and


Melesse, 2012;Mwangi et al., 2016), in this study water level time series are used to avoid the uncertainties in
the discharge data.

During field trips, some point discharge measurements were done in September/October 2014 at Emarti Bridge,
Serena Pump House and New Mara Bridge, see Table 2 and Fig. 2. At each location, the discharge was derived

from cross-section and velocity measurements done with a RiverSurveyor, a small boat that was pulled across
the river and on which was mounted an Acoustic Doppler Profiler, a Power Communications Module and a
DGPS antenna (Rey et al., 2015).

**Table 1: Hydro-meteorological data availability in the Mara River Basin. The temporal coverage for water level and**
**discharge can be different due to poor administration.**

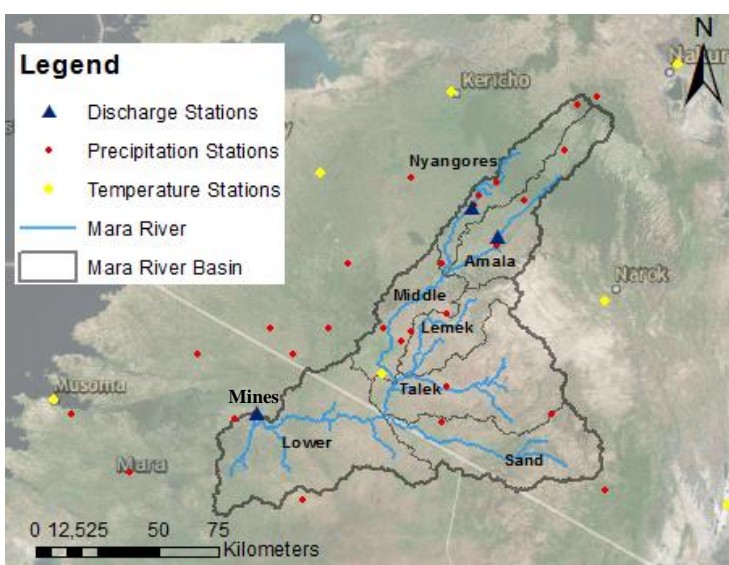

**Figure 1: Map of the Mara River Basin and the hydro-meteorological stations for which data is available**

**Table 2: Discharge measured in the field using a RiverSurveyor at three locations in the Mara River Basin. A
RiverSurveyor is a small boat on which an Acoustic Doppler Profiler, a Power Communications Module and a DGPS
antenna was mounted (Rey et al., 2015)**

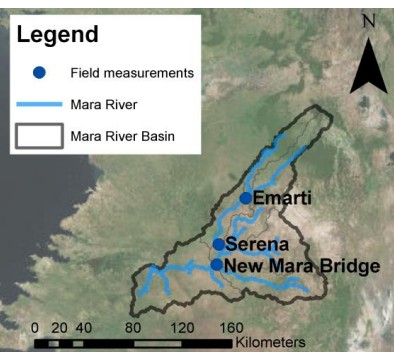

**Figure 2: Map of discharge measurement locations during field trips in September/October 2014**



### 3 Hydrological model setup for the Mara River Basin

#### 3.1 Catchment classification based on landscape and land use

For this study, the modelling concept of FLEX-Topo was used. It is a semi-distributed rainfall runoff modelling framework that distinguishes hydrological response units (HRUs) based on landscape features. The landscapes classes were identified based on the topographical indexes HAND (Height Above Nearest Drain) and slope (Savenije, 2010) using a digital elevation map (SRTM) with a resolution of 90 m and vertical accuracy of 16 m (U.S. Geological Survey, 2014). Hillslopes are defined by a strong slope (more than 12.9%) and high HAND

(more than 5.9 m); wetlands by a low HAND; and terraces by a high HAND and mild slope. The thresholds for the slope and HAND were based on a sensitivity analyses within the Mara Basin. In the Mara River Basin, there are mainly terraces and hill slopes. To further delimit HRUs, land cover is taken into account based on Africover, a land cover database based on ground truth and satellite images (FAO, 1998). This resulted in four HRUs in the sub-basin of the Mara River Basin: forested hill slopes, shrubs on hill slopes, agriculture and

grassland (Fig. 3, Fig. 4 and Table 3). In the upper sub-catchments, there are mainly cropland and forest, whereas further south the land use is dominated by grassland. In the lower sub-catchment, there is mostly cropland.

**Table 3: Classification results: area percentage of each hydrological response unit per sub-catchment in the Mara River Basin**

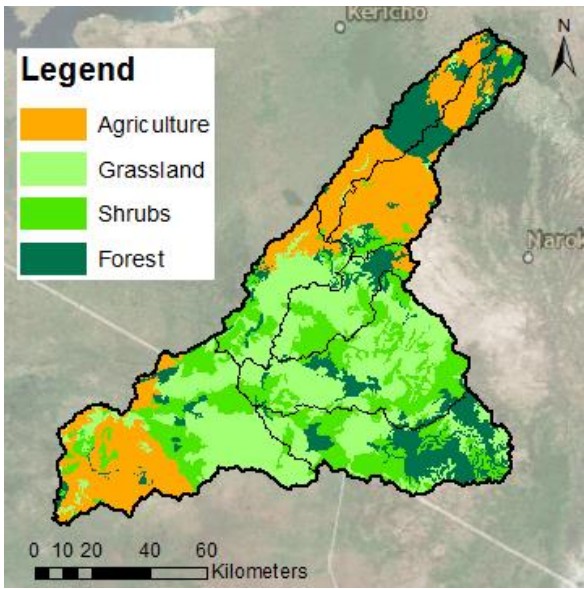

**Figure 3: Classification of the Mara River Basin into four hydrological response units for each sub-catchment based on land use and landscape**



## 3.2 Hydrological model structure

Each HRU is represented by a lumped conceptual model; the model structure is based on the dominant flow processes observed during field trips or deducted from interviews with local people. For example in forests and shrub lands, Shallow Subsurface Flow (SSF) was seen to be the dominating flow mechanism: Rainwater infiltrates into the soil and flows through preferential flow paths to the river. In contrast, grassland and cropland generate overland flow. The observed soil compaction, due to cattle trampling and ploughing, reduces the preferential infiltration capacity resulting in overland flow during heavy rainfall. Consequently, in these land Hortonian Overland Flow (HOF) occurs at high rainfall intensities excessing the maximum infiltration capacity. The perception of the dominant flow mechanisms (Fig. 4) was then used to identify a suitable model structure (Fig. 5). This approach of translating a perceptual model into a model concept (Beven, 2012) was applied successfully in previous FLEX-Topo applications (Gao et al., 2014a;Gharari et al., 2015).

The model structure contains multiple storage components schematised as reservoirs (Fig. 5). For each reservoir, the inflow, outflow and storage are defined by water balance equations, see Table 4. Process equations determine the fluxes between these reservoirs as a function of input drivers and their storage. HRUs function in parallel and independently from each other. However, they are connected through the groundwater system and the drainage network. To find the total runoff at the sub-catchment outlet $Q_{m,sub}$, the outflow $Q_{m,i}$ of each HRU is multiplied by its area percentage and then added up together with the groundwater discharge $Q_s$. The area percentage is the area of a specific HRU divided by the entire sub-catchment area. Subsequently, the modelled discharge at the catchment outlet is obtained by using a simple river routing technique where a delay from sub-catchment outlet to catchment outlet was added assuming an average river flow velocity of 0.5 m/s. In the Sand sub-catchment, it is schematised that runoff can percolate to the groundwater from the river bed and that moisture can evaporate from the groundwater through deep rooting or riparian vegetation.

**Table 4: Equations applied in the hydrological model. The formulas for the unsaturated zone are written for the hydrological response units: Forested hill slopes and Shrubs on hill slopes; for grass and agriculture, the inflow Pe changes to QF. The modelling time step is $\Delta t = 1$ day. Note that at a time daily step, the transfer of interception storage between consecutive days is assumed to be negligible.**





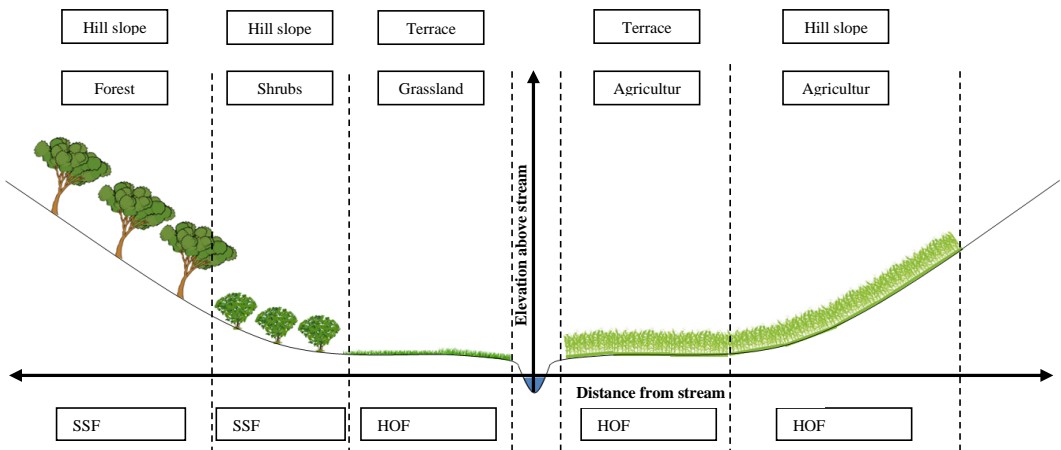

**Figure 4: Schematization of the landscape and land use based classification**

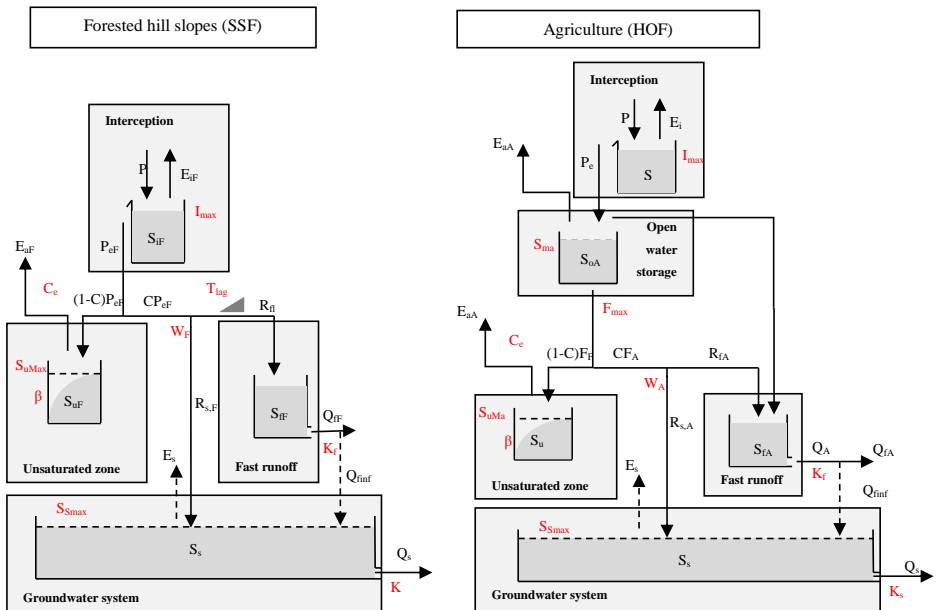

**Figure 5: Model structure of the HRUs: Forested hill slopes (left) and Agriculture (right). The structure for Shrubs on hill slopes is similar to the left one replacing the indices F with S. The structure for Grassland is similar to the right one replacing the indices A with G. Parameters are marked in red, storages and fluxed in black. Symbol explanation: Fluxes: precipitation (P), evaporation of the interception zone (Ei), actual evaporation (Ea), evaporation from groundwater only applied in the sub-catchment Sand (Es), effective precipitation (Pe), infiltration into the unsaturated zone (FA), discharge from unsaturated zone to the fast runoff zone (Rf), groundwater recharge (Rs), discharge from the fast runoff (Qf), infiltration into groundwater system only applied in the sub-catchment Sand (Qf, inf), discharge from the slow runoff (Qs). Storages: storage in the interception zone (Si), open water storage (SoA), storage in the root zone (Su), storage for the slow runoff (Ss), storage for the fast runoff (Sf). Remaining symbols: splitter (W), splitter (C), soil moisture distribution coefficient (β), transpiration coefficient (Ce = 0.5), reservoir coefficient (K); indices f and s indicate the fast and slow runoff. Units: fluxes [mm/d], storages [mm], reservoir coefficient [d], remaining parameters [-].**





### 3.3 Calibration and validation strategy using water level data

Parameters and process constraints have been applied to eliminate unrealistic model results and constrain the flow volume. For example, the maximum storage in the unsaturated zone $S_{u,max}$, equal to the root zone storage

capacity, has been estimated based on the method of Gao (2014) using remote sensed precipitation and evaporation data (Gao et al., 2014b;Wang-Erlandsson et al., 2016). The dry season evaporation has been derived from the actual evaporation using the NDVI. In addition, the total evaporation has been constrained using the Budyko curve (Gharari et al., 2015). In the supplement, a list of all parameter and process constraints is presented including their equations and graphs illustrating the influence of the process constraints.


After having set up the model and defined the constraints, the model was calibrated and evaluated. The hydrological model was calibrated on water levels due to lack of reliable discharge data. For the evaluation of this calibration, the Nash Sutcliffe coefficient was used on the flow duration curve and its logarithm, see Eq. (1). The modelled water depth $d_{mod}$ was calculated from the modelled discharge $Q_{mod}$ using the Strickler formula and

the cross-sectional geometry ($Q = k * i^{\frac{1}{2}} * A * R^{\frac{2}{3}} = c * A * R^{\frac{2}{3}}$), where R is the hydraulic radius and A the cross-sectional area; the unknown parameter c was calibrated. Note that by using the Strickler formula the exponent of the rating curve is fixed; $Q = a * (h - h_0)^b$. Also note that the parameter c compensates for non-closure of the water balance. Therefore the calibrated c values have to be checked whether they are in a feasible range of roughness and slope values. Subsequently, the discharge was estimated with the same Strickler formula, but

now using the observed water depth $d_{obs}$ which is the water level subtracted by the reference level. This discharge $Q_{Strickler}$ was then compared to the modelled discharge $Q_{mod}$ and the recorded discharge $Q_{rec}$. As a result new geometric rating curves were obtained (relation between $Q_{Strickler}$ and $d_{obs}$) and compared to the recorded rating curves (Table 5 for a schematisation of the methodology).

The model was run for the entire catchment using the station Mines, and for the sub-catchments Nyangores and Amala. For each simulation, the obtained water depth was evaluated by the flow duration curve, the water level time series and the logarithm of the time series. The selected time periods for each simulation were:

Mines (5H2):

-        Calibration        1970-1974

-        Validation 1        1980-1981

-        Validation 2        1982-1983

Nyangores (1LA03):

-        Calibration        1970-1980

-        Validation        1981-1992

Amala (1LB02):

-        Calibration        1991-1992

-        Validation        1985-1986





**Equation 1: Formulas for the Nash-Sutcliff objective function. The indices mod and obs indicate modelled and observed values, respectively. In all cases, sorted data was used for the calculation of the objective function therefore the flow duration curve was calibrated.**

$$NS_{log(d)} = 1 - \frac{\Sigma\big(log(d_{mod,sorted}) - log(d_{obs,sorted})\big)}{\Sigma\big(log(d_{obs,sorted}) - log(d_{obs,avg})\big)} \qquad NS_d = 1 - \frac{\Sigma\big(d_{mod,sorted} - d_{obs,sorted}\big)}{\Sigma\big(d_{obs,sorted} - d_{obs,avg}\big)}$$

**Table 5: Schematisation of the methodology**

**3.4 Precipitation input data**

For the precipitation input data, a single station was chosen for each sub-catchment assuming it was representative for the entire area (Fig. 6 A). However, the representative average precipitation in an sub-catchment can also be estimated using Thiessen polygons (Fig. 6 B). Alternatively, multiple precipitation stations can be used within a single sub-catchment by splitting it up into different areas with the same

precipitation based on Thiessen polygons. Therefore, the following three methods were used to estimate the representative precipitation for the hydrological model:

- Method 1: Selection of a single station for each sub-catchment assuming it was representative for the entire area
- Method 2: Calculation of the representative average precipitation for each sub-catchment using Thiessen

polygons
- Method 3: Sub-division of each sub-catchment into areas with equal rainfall using Thiessen polygons

Method 1 was used as reference and the remaining two methods to assess the model sensitivity to areal rainfall estimates.

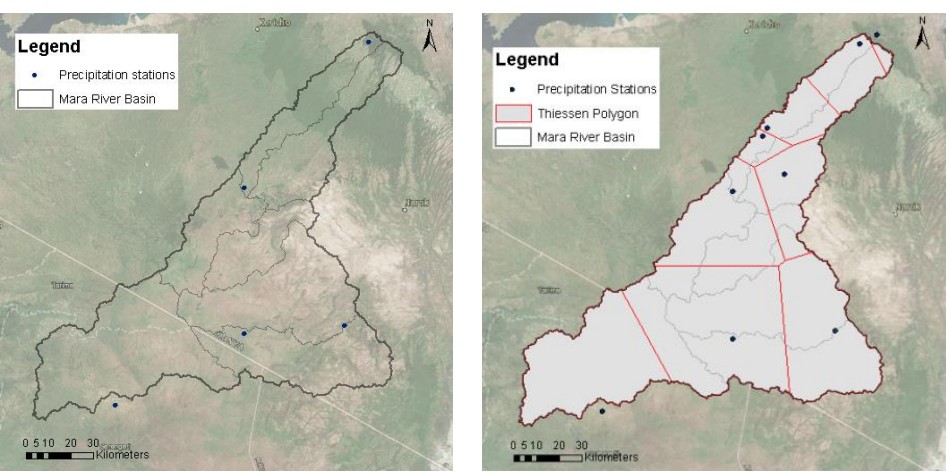


**Figure 6: Map of the precipitation stations used for modelling based on A) Method 1 and B) Method 2 and 3 for areal rainfall estimates. Method 1: Single precipitation station for each sub-catchment; Method 2: Representative average precipitation for each sub-catchment using Thiessen polygons; Method 3: Sub-division of each sub-catchment into areas with equal rainfall using Thiessen polygons**





## 4 Results and discussion

### 4.1 Water depth and flow duration curve

The results of the objective functions indicate that at Nyangores and Mines the validation results were more consistent (Table 6). At Mines, the observed and modelled water depth were quite similar to each other,

particularly with regard to the duration curve (Table 6 and Fig. 7). At individual events, there were substantial differences, but this could be due to the spatial heterogeneity of the rainfall that were not represented well in the forcing data. On the other hand, the year 1974 was well represented. In general, the model captured the dynamics in the water depth well. This was the case during both calibration and validation (see supplement).

At Nyangores the observed and modelled water depths were also similar during calibration and validation, extreme high flows excluded (Fig. 8). However at Amala, the observed and modelled water depths differed significantly during calibration (Fig. 9) and validation. The model missed several rain events completely, likely linked to the high heterogeneity in precipitation. Also there seemed to be backwater effects raising the water level, possibly due to a river blockage such as a weir, sand dam or dunes.

**Table 6: Overview of the values of the objective functions for each model simulation. Calibration was done based on the water depth: $NS_{log(d)}$ and $NS_d$; for comparison, objective functions using the discharge were added here as well**

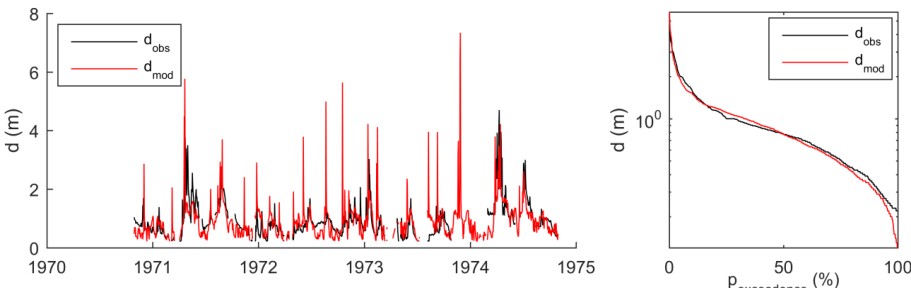

**Figure 7: Model results at Mines during calibration: water depth time series and water depth exceedance**

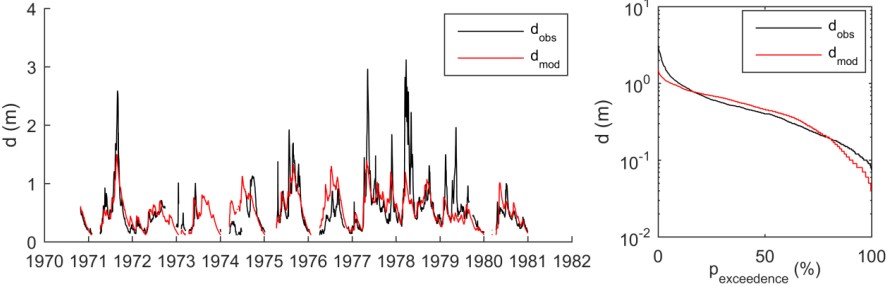

**Figure 8: Model results at Nyangores during calibration: water depth time series and water depth exceedance**



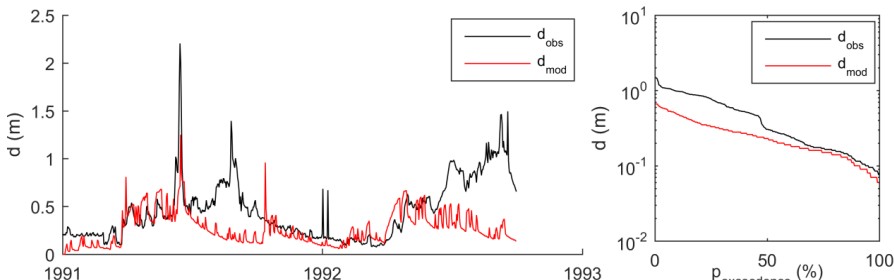

**Figure 9: Model results at Amala during calibration: water depth time series and water depth exceedance**


**4.2 Discharge at sub-catchment level**

At Mines, the discharge originates from seven different sub-catchments, each with a different contribution. Based on field observations, the upper sub-catchments should have the largest contribution whereas the contribution from the relatively drier and flatter Lemek and Talek should be relatively low. The contribution of 285 each sub-catchment to the total modelled discharge was assessed on a monthly timescale and compared with observations.

As shown in Fig. 10, the contribution varied throughout the year. In the summer (July-September), the modelled discharge mainly originates from the upper sub-catchments, Nyangores and Amala, just as expected. However 290 in the winter (November-April), the modelled discharge mainly originates from the Sand and Lower sub-catchment. The Middle, Talek and Lemek sub-catchments have the lowest discharge throughout the entire year just as observed.

**Figure 10: Monthly averaged modelled discharge for each sub-catchment**

To validate the model at sub-catchment level, model results were compared with discharge measurements done during field trips in September/October 2014 at Emarti Bridge, Serena Pump House and New Mara Bridge. At all three locations, the modelled discharge in the same month was of the same order of magnitude as the point measurement, see Fig. 11. In previous studies, it was shown that only a few discharge measurements contain sufficient information to constrain model predictive uncertainties effectively (Seibert and Beven, 2009).





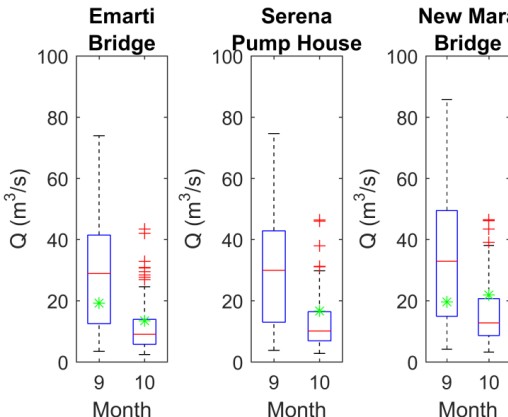


**Figure 11: Boxplot of the modelled discharge at three locations; the green asterix represents the measured discharge in Sep/Oct 2014**

**4.3 Rating curve analysis**

The discharge and rating curves have been evaluated by analysing discharge - water depth graphs which basically plot the rating curve. In this study, two different rating curves are distinguished:

- "Geometric rating curve", relating $Q_{Strickler}$ to $d_{obs}$, and
- "Recorded rating curve", relating $Q_{rec}$ to $d_{obs}$

At Mines, the modelled discharge correlated with $Q_{Strickler}$, but with considerable scatter (Fig. 12). Comparison 310 of the recorded discharge and $Q_{Strickler}$ however, revealed that the recorded discharge was lower. Therefore also the recorded and geometric rating curves were significantly different from each other. However, for high flows both rating curves, recorded and geometric, were parallel to each other indicating similar cross-sectional properties. This observation reoccurred during validation as well.

The significant difference between the recorded and geometric rating curve at Mines can be a result of uncertainties in the available recorded discharge data, hence the recorded rating curve. In the discharge - water depth graphs at Mines (see supplement), a large scatter is found in the observation which should not be the case assuming one rating curve was used, compared with Nyangores where there is no scatter. This scatter could be the result of variability in for example the reference water level $h_0$ in the rating curve equation for example due 320 to sand banks and bed forms. A sensitivity analysis of the recorded rating curve equation at Mines showed that a deviation 0.1 m in the reference water level altered the discharge with 4% - 46%, lowest for high flows and highest for low flows. However, a deviation of 0.5 m in the reference water level resulted in a 19% - 325% change in the discharge. Therefore, variability in the reference water level increases the uncertainty in the recorded rating curve. The uncertainties in the discharge data can also be seen in the calculated cross-section 325 average flow velocity based on the recorded discharge and water level data: this was below 1 m/s (see supplement) whereas for example the measured velocity in 2012 was 2.13 m/s (GLOWS-FIU, 2012).





At Nyangores and Amala, the modelled discharge correlated with $Q_{Strickler}$, also with considerable scatter (Fig. 12). The recorded and geometric rating curves were almost identical at Nyangores, but not at Amala.

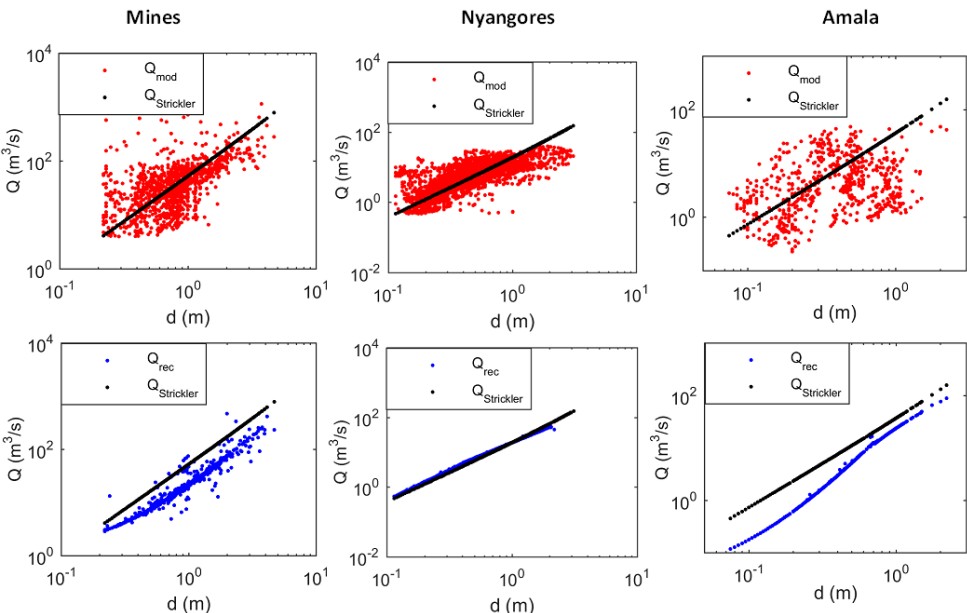

**Figure 12: Model calibration results at Mines, Nyangores and Amala: Discharge - water depth graphs**

### 4.4 Sensitivity to areal rainfall estimates

In the previous sections, it was shown that water level data can be used instead of discharge data to calibrate a model and to establish a rating curve equation. However, how sensitive is this method to areal rainfall estimation methodologies? This was analysed by comparing three different methods of representative rainfall estimates for each sub-catchment: 1) single station, 2) average of multiple stations based on Thiessen polygons, 3) sub-division into areas with equal rainfall based on Thiessen polygons. All three methods resulted in different daily or monthly rainfall values; the maximum difference was 86 mm/month at Amala in August (Fig. 13). In general, there were more dry days when using a single station for each sub-catchment (method 1). Also, when using Thiessen polygons (methods 2 and 3), rainfall events were more dampened as a result of averaging multiple stations.

These differences in the precipitation data were reflected in the modelled water depth. Compared to the observation, method 1 resulted in very flashy responses and method 2 very dampened ones whereas method 3 was a combination of both (Fig. 14). The change in precipitation input data also influenced the geometric rating curve as shown in Table 7: the constant, parameter a, in the rating curve equation $Q = a * (h - h_0)^b$ increased with 45% and 35% for methods 2 and 3 respectively. This difference was within the modelling uncertainty bounds which was 75% in this case. However, this change in the rating curve constant indicates that the model compensated errors in the rainfall data by closing the water balance.





Besides altering the geometric rating curve equation, the precipitation estimation method also influenced the modelled annual averaged runoff coefficient (Fig. 15). Averaged over the entire river catchment, this difference

in runoff coefficient was insignificant, however on sub-catchment level, the largest variation was found in the Sand sub-catchment: the runoff coefficient changed from 5% with method 1 to 1% with method 2.

**Table 7: Recorded rating curve and model results for the geometric rating curves using three different methods for areal rainfall estimates. Method 1: Single precipitation station for each sub-catchment; Method 2: Representative**
**average precipitation for each sub-catchment using Thiessen polygons; Method 3: Sub-division of each sub-catchment into areas with equal rainfall using Thiessen polygons**

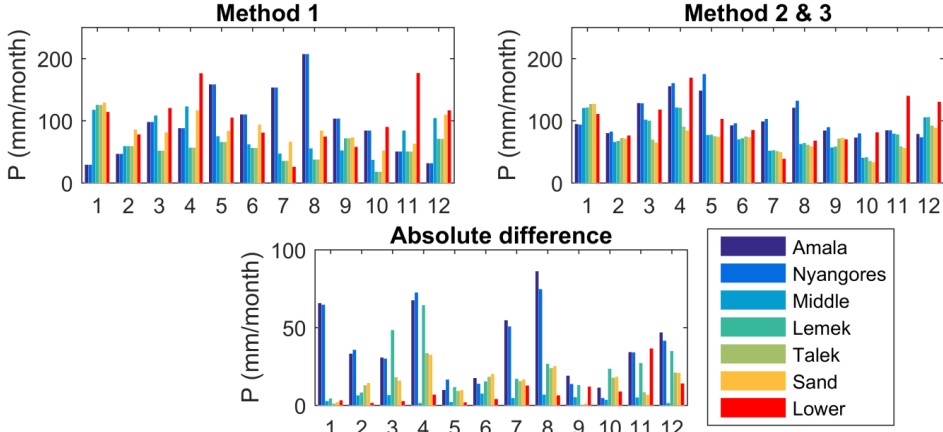

**Figure 13: Monthly average precipitation per sub-catchment. A) Method 1, B) Method 2 and 3, C) Absolute difference. Method 1: Single precipitation station for each sub-catchment; Method 2: Representative average precipitation for each sub-catchment using Thiessen polygons; Method 3: Sub-division of each sub-catchment into areas with equal rainfall using Thiessen polygons**





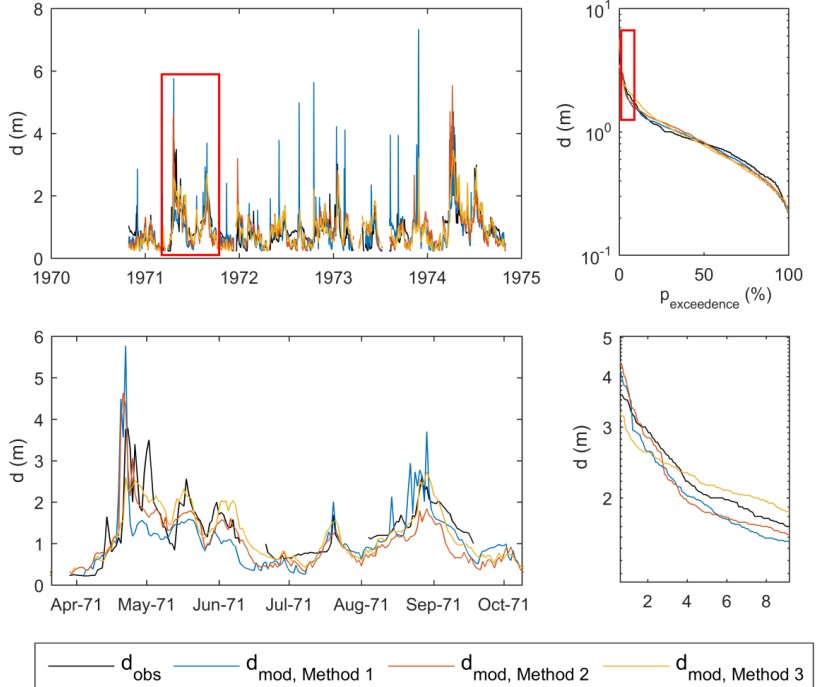


**Figure 14: Modelled water depth for the Mara River Basin at Mines: time series (left) and flow duration curve (right) for the entire modelled time series (upper) and zoomed in a section marked in the red boxes (lower) using the model parameters obtained with three methods for areal rainfall estimates. Method 1: Single precipitation station for each sub-catchment; Method 2: Representative average precipitation for each sub-catchment using Thiessen polygons;**
**Method 3: Sub-division of each sub-catchment into areas with equal rainfall using Thiessen polygons**

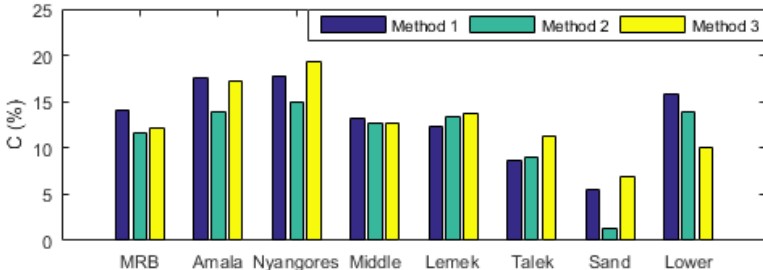

**Figure 15: Modelled runoff coefficient for the entire Mara River Basin (MRB) and each sub-catchment with the three methods for areal rainfall estimates. Method 1: Single precipitation station for each sub-catchment; Method 2:**
**Representative average precipitation for each sub-catchment using Thiessen polygons; Method 3: Sub-division of each sub-catchment into areas with equal rainfall using Thiessen polygons**





**5 Conclusion**

Hydrological models play an important role in Water Resources Management. Unfortunately, the quantity and
quality of the available discharge measurements are often inadequate for reliable hydrological modelling in
African river catchments. There are various sources of uncertainty in discharge time series when using rating
curves due to extrapolation to estimate flood peaks or non-stationarity due to sedimentation or erosion altering
the cross-section. To cope with these uncertainties during model calibrations, there are two options: 1) assess the
uncertainty in discharge data and its effect on model predictions, or 2) avoid these uncertainties by using water
level data instead.

In this study, a hydrological model is developed for the semi-arid and poorly gauged Mara River Basin as a case
study. The effects of the discharge data uncertainties are avoided by using water level instead of discharge time
series by incorporating the hydraulic equation describing the rating curve within the model. A semi-distributed
rainfall runoff modelling framework called FLEX-Topo was used to model the Mara River Basin. The
catchment was split into four hydrological response units (HRUs) and seven sub-catchments based on the river
tributaries. For each HRU, a unique model structure was defined based on the expected dominant flow
processes. By constraining the parameters and processes, unrealistic results were excluded from the calibration
parameter set and the flow volume was constrained. This model was then calibrated based on water depths to
capture the flow dynamics; modelled water depths were calculated from modelled discharges with cross-section
data and the Strickler formula.

The hydrological model simulated the water depths well for the entire basin and the Nyangores sub-catchment
in the north. In addition, a new geometric rating curve was calibrated based on the modelled discharge, observed
water level and the Strickler formula. The geometric and recorded rating curve were slightly different at Mines,
the catchment outlet, probably due to uncertainties in the recorded discharge data. At Nyangores however, the
modelled and recorded discharge were almost identical. In addition, it was found that the precipitation
estimation methodology influenced the model results significantly: application of a single station for each sub-
catchment resulted in flashier responses whereas Thiessen averaged precipitation resulted in more dampened
responses. The inadequate knowledge of the spatial distribution of the precipitation was the main limitation for
accurate rainfall-runoff modelling. Therefore rapidly improving precipitation monitoring methods from space
offer promising approximations for improving rainfall-runoff modelling in poorly gauged basins. Note that by
calibrating the unknown parameter of the hydraulic equation, a combination of slope and roughness, the non-
closure of the water balance is compensated as also errors in the rainfall data. Therefore, calibrated parameter
values should be verified and if possible constrained.

In conclusion, promising results have been obtained when using water level time series for calibrating the
hydrological model of the Mara River Basin in combination with process controls to constrain the flow volume.

**Acknowledgement**
This research was part of the MaMaSe project (Mau Mara Serengeti) led by IHE Delft. Station data (discharge,
water level and precipitation) was provided by the Water Resource Management Authority (WRMA) in Kenya.
Temperature and additional precipitation data was obtained from NOAA online database (Menne et al., 2012).





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





**Table 1: Hydro-meteorological data availability in the Mara River Basin. The temporal coverage for water level and discharge can be different due to poor administration.**


|  | Precipitation | Temperature | Water level, discharge | | |
|---|---|---|---|---|---|
| **Number of stations** | 29 | 5 | 3 | | |
| **Station ID** | - | - | 1LA03 | 1LB02 | 5H2 |
| **Station location** | - | - | Nyangores at Bomet | Amala at Kapkimolwa | Mara at Mines |
| **Time range** | 1959 -2011 | 1957 - 2014 | 1963-2009 | 1955-2015 | 1969-2013 |
| **Duration [years]** | 0 - 43 | 3 - 57 | 46 | 60 | 44 |
| **Coverage** | 8 - 100% | 30 -100% | Discharge: 85% Water level: 85% | Discharge: 72% Water level: 70% | Discharge: 53% Water level: 61% |

**Table 2: Discharge measured in the field using a RiverSurveyor at three locations in the Mara River Basin. A RiverSurveyor is a small boat on which an Acoustic Doppler Profiler, a Power Communications Module and a DGPS antenna was mounted (Rey et al., 2015)**

| Station name | Date | Mean discharge | Standard deviation |
|---|---|---|---|
| **Emarti Bridge** | 13 Sep 2014 | 19.2 $m^3/s$ | 0.7 $m^3/s$ |
|  | 4 Oct 2014 | 13.4 $m^3/s$ | 0.6 $m^3/s$ |
| **Serena Pump House** | 9 Oct 2014 | 16.6 $m^3/s$ | 0.4 $m^3/s$ |
| **New Mara Bridge** | 19 Sep 2014 | 19.6 $m^3/s$ | 0.6 $m^3/s$ |
|  | 6 Oct 2014 | 21.9 $m^3/s$ | 0.4 $m^3/s$ |


**Table 3: Classification results: area percentage of each hydrological response unit per sub-catchment in the Mara River Basin**

| Sub-catchment | Agri-culture | Shrubs on hill slopes | Grassl and | Forested hill slopes |
|---|---|---|---|---|
| **Amala** | 67% | 0% | 0% | 33% |
| **Nyangores** | 61% | 0% | 0% | 39% |
| **Middle** | 19% | 16% | 65% | 0% |
| **Lemek** | 10% | 39% | 51% | 0% |
| **Talek** | 0% | 21% | 79% | 0% |
| **Sand** | 0% | 42% | 58% | 0% |
| **Lower** | 26% | 23% | 52% | 0% |





**Table 4: Equations applied in the hydrological model. The formulas for the unsaturated zone are written for the hydrological response units: Forested hill slopes and Shrubs on hill slopes; for grass and agriculture, the inflow $P_e$ changes to $Q_F$. The modelling time step is $\Delta t = 1$ day. Note that at a time daily step, the transfer of interception storage between consecutive days is assumed to be negligible.**

| Reservoir system | Water balance equation | Process functions |
|---|---|---|
| **Interception** | $\frac{\Delta S_i}{\Delta t} = P - P_e - E_i \approx 0$ | $E_i = \min\left(E_p, \min\left(P, \frac{I_{max}}{\Delta t}\right)\right)$ |
| **Surface** | $\frac{\Delta S_o}{\Delta t} = P_e - Q_F - Q_{HOF} - E_o$ | $Q_F = \min(\frac{S_o}{\Delta t}, F_{max})$ |
| | | $Q_{HOF} = \max\frac{(0, S_o - S_{max})}{\Delta t}$ |
| | | $E_o = \max(0, \min\left(E_p - E_i, \frac{S_o}{\Delta t}\right))$ |
| **Unsaturated zone** | $\frac{\Delta S_u}{\Delta t} = (1 - C) * P_e - E$ | $C = 1 - \left(1 - \frac{S_u}{S_{u,max}}\right)^\beta$ |
| | | $E = \min((E_p - E_i), \min\left(\frac{S_u}{\Delta t}, (E_p - E_i) * \frac{S_u}{S_{u,max}} * \frac{1}{C_e}\right))$ |
| **Groundwater recharge** | | $R_s = W * C * P_e$ |
| **Fast runoff** | $\frac{\Delta S_f}{\Delta t} = R_{fl} - Q_f$ | $R_{fl} = T_{lag}(C * P_e - R_s)$ → in a linear delay function $T_{lag}$ |
| | | $Q_f = \frac{S_f}{K_f}$ |
| **Groundwater** | $\frac{\Delta S_s}{\Delta t} = R_{s,tot} - Q_s - E_s + Q_{inf}$ | $R_{s,tot} = \sum_{i=1}^{i=4} R_{s;HRU_i}$ |
| | | $Q_s = \frac{S_s}{K_s}$ |
| | | $E_s = 0$ and $Q_{inf} = 0$ for all sub − basins except Sand |
| | | $Q_{inf} = \min\left(\frac{S_{s,max} - S_s}{\Delta t}, Q_f\right)$ for Sand sub − basin |
| | | $E_s = \max\left(0, \min\left(E_p - E_i - E_o - E, \frac{S_s}{\Delta t}\right)\right)$ for Sand sub − basin |
| **Total runoff** | | $Q_m = Q_s + \sum_{i=1}^{i=4} Q_{f;HRU_i}$ |

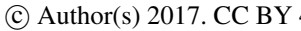




**Table 5: Schematisation of the methodology**

| | | Model input data | Model output data |
|---|---|---|---|
| **Model** | Precipitation Temperature | FLEX-Topo | $Q_{mod}$ (Modelled discharge, output from FLEX-Topo) |
| | | | $Q_{Strickler}$ (Discharge calculated with $d_{obs}$ using the Strickler formula, parameter c is calibrated) |
| | | | Strickler → $d_{mod}$ (Modelled water depth calculated with $Q_{mod}$ using the Strickler formula, parameter c is calibrated) |
| | | | Geometric rating curve ⇄ Calibration |
| **Observation** | | | Recorded rating curve → $d_{obs}$ (Observed water depth) |
| | | | $Q_{rec}$ (Recorded discharge) |





**Table 6: Overview of the values of the objective functions for each model simulation. Calibration was done based on the water depth: $NS_{log(d)}$ and $NS_d$; for comparison, objective functions using the discharge were added here as well**

| | Nyangores | | Amala | | Mines | | |
| --- | --- | --- | --- | --- | --- | --- | --- |
| | Calibration | Validation | Calibration | Validation | Calibration | Validation 1 | Validation 2 |
| $NS_{log(d)}$ | 0.92 | 0.75 | 0.92 | -0.23 | 0.97 | 0.81 | 0.93 |
| $NS_d$ | 0.80 | 0.69 | 0.26 | 0.37 | 0.97 | 0.92 | 0.89 |
| $NS_{log(Q)}$ | 0.92 | 0.69 | 0.57 | 0.63 | 0.97 | 0.81 | 0.93 |
| $NS_Q$ | 0.55 | 0.37 | 0.08 | -1.67 | 0.90 | 0.76 | 0.77 |


**Table 7: Recorded rating curve and model results for the geometric rating curves using three different methods for areal rainfall estimates. Method 1: Single precipitation station for each sub-catchment; Method 2: Representative average precipitation for each sub-catchment using Thiessen polygons; Method 3: Sub-division of each sub-catchment into areas with equal rainfall using Thiessen polygons**

| | Method 1 | Method 2 | Method 3 |
| --- | --- | --- | --- |
| **Recorded rating curve** | $Q_{obs} = 23.1 * (h - h_0)^{1.54}$ | $Q_{obs} = 23.1 * (h - h_0)^{1.54}$ | $Q_{obs} = 23.1 * (h - h_0)^{1.54}$ |
| **Geometric rating curve** | $Q_{Str} = 52.5 * (h - h_0)^{1.70}$ | $Q_{Str} = 47.4 * (h - h_0)^{1.70}$ | $Q_{Str} = 46.3 * (h - h_0)^{1.70}$ |
