# Peer review of "Rainfall-runoff modelling using river stage time series in the absence of reliable discharge information: a case study in the semi-arid Mara River Basin"

_Hydrology and Earth System Sciences, 2017_

## Short Comment (SC1) · 11 Dec 2017

Review of "Modelling the Mara River Basin with data uncertainty using water levels for calibration" By Petra Hulsman, Thom A. Bogaard, Hubert H.G. Savenije

The manuscript describes an approach to reduce the effect of discharge uncertainty in the calibration process of hydrological models. The authors suggest using water level observations instead of discharge data to evade the uncertainties related to rating curves. They have calibrated FLEX-Topo model to water level observations at three gauges of Mara watershed located in Kenya and Tanzania. Overall, the research is interesting; however, I have several major and minor concerns regarding the manuscript. I hope the authors find my comments helpful as summarized in the following.
Major concerns:

1. As far as I have understood, the authors have used water level observations (dobs) to calibrate the model. So, Manning-Strickler formula has been implemented in the model (line 60) to simulate both discharge, Qmod, and water levels, dmod. Moreover, the authors have produced discharges based on dobs using Manning-Strickler formula and named it QStrickler. Then they compared the recorded observed discharge, Qrec, with QStrickler and Qmod in Figure 12. How did the authors produce QStrickler? Have you had information about the cross-section details at three locations indicated in Figure 12? The research method explanation is hard to follow and understand.

2. The explanation of the methodology used in the manuscript is vague. Table 5 can be improved significantly.

3. There is no information about the calibration process of either the FLEX-Topo model or the Manning-Strickler formula. What were the initial ranges of parameters? How many parameters have been calibrated? Did the authors used an optimization algorithm or an uncertainty-based method? What were the final ranges/values of parameters? Have you tried any other objective function rather than Nash-Sutcliffe? Why have the authors used two validation periods for Mines (lines 221-222)?

4. The time-step of the model seems to be neglected. The information about the time-step is not discussed in the paper expect a minor reference under Table 4 caption. Have you tried different time-steps? Could results improve if you use a smaller time-step?

5. One of the main purposes of hydrological models is producing the hydrographs at different locations. Although authors have tried to indicate the water level time series (Figures 7, 8, 9 and 14), the hydrographs are missing.

6. The details of sensitivity analysis to produce thresholds of different landscape slopes and HAND values are missing. Is the HAND model based on the research of Nobre et al. (2011)? Have you used any specific sensitivity analysis algorithm/approach?

7. Are calibrated roughness values in accordance with the streambed material for Manning-Strickler formula?

8. How did the authors specify the average flow velocity (line 165)? Would changing this parameter value impact the overall results? Does it change the hypothesis of using Manning-Strickler formula?

Minor concerns:

1. The use of English language should be improved significantly. A number of grammatical errors could be found (e.g., in line 237 "an sub-catchment") and also some sentences are not clear and easy to understand (e.g., "a large scatter is found in the observations which could not be the case assuming one rating curve was used." (lines 317-318), "... the parameter c compensates for non-closure of the water balance" (lines 207-208) and etc.)

2. The title of research seems awkward. What does 'modeling [. . .] with data uncertainty' mean? Where did the uncertainty of streamflow, either water level or discharge, come into consideration?

3. I do suggest a separate section for data as different data sources have been mentioned in different places (e.g., field trip data (lines 108 to 112), digital elevation map (line 128), Africover database (line 133), precipitation and evaporation data (lines 195-196, line 423), etc.)

4. Some of the mentioned sources are not available in the reference list. For instance, Karamuz et al. (2016) (lines 57-58), GLOWS-FIU (2012) (line 326). Also, some of the reference details are flawed, such as the reference to the paper by Gharari et al. (2014) which is wrong. The paper is published in 2014, not 2015 (line 453).

5. Equation 1 indicating Nash-Sutcliffe formula is wrong (lines 230 to 233).

6. In several figures, there are plenty of sub-figures (Figures 5, 6, 11, 12, 13,14 and several ones in supplement) which could be denoted by letters or numbers to avoid

confusions. In Figures 1 and 2, the sub-catchment boundary lines are not introduced in figure legends. Moreover, in Figures 10, 11 and 13 the number of months could be replaced by their actual names.

7. What is the time period of discharge data indicated in Figure 12?

8. The 'Strickler' formula is also known as 'Manning' equation/formula. Authors could have used the 'Manning-Strickler' term which is more general.

9. Instead of using the phrase "see supplement" in multiple locations the authors could refer to the specific figure or table in the supplements. For instance, in line 198, they could have specifically referred to Table S1 and Table S2.

10. The number of temperature stations is different as declared in line 89 compared to Figure 1 and supplement spreadsheet data.

11. Section 4.4 needs more discussion as no general suggestion to future research is made. Moreover, it is not apparent whether these strategies have improved the results of calibration.

12. The conclusions need to be considered again as many ideas have been repeated from the introduction/abstract part. It could have been more concise and explicit.

Given the major and minor comments provided in this review, the manuscript should be improved significantly in my point of view to meet the minimum levels of quality for publication in HESS.

With kind regards,

Kasra Keshavarz

PS. I would like to thank Dr. Jeffrey McDonnell and Dr. Shervan Gharari who helped in reviewing this paper.

—-

References:

1. Nobre, A. D., Cuartas, L. A., Hodnett, M., Rennó, C. D., Rodrigues, G., Silveira, A., Waterloo, M. and Saleska, S.: Height Above the Nearest Drainage - a hydrologically relevant new terrain model, J. Hydrol., 404(1–2), 13–29, doi:10.1016/j.jhydrol.2011.03.051, 2011.

2. Gharari, S., Hrachowitz, M., Fenicia, F., Gao, H. and Savenije, H. H. G.: Using expert knowledge to increase realism in environmental system models can dramatically reduce the need for calibration, Hydrol. Earth Syst. Sci., 18(12), 4839–4859, doi:10.5194/hess-18-4839-2014, 2014.

---

## Referee Comment (RC1) · Anonymous Referee #1 · 20 Dec 2017

The objective of this paper is to build a model for the Mara basin in Kenya. I found the paper potentially interesting in terms of the methodologies it introduces for dealing poorly gauged catchments. However, it needs major improvements. Below my detailed comments.

Section 1. The paper states that the "The goal is to develop a reliable hydrological model for the semi-arid and poorly gauged Mara". In my opinion, this is not the kind of objectives that warrants a publication. I am convinced that the authors can identify a set of more appealing objectives for their work.

Section 1. Can the authors clarify why using water levels for model calibration avoids the effect of discharge uncertainties? This is presented as a fact, with no references to previous literature, and no explanations. I do not find the explanation obvious. Do they

imply that rating curves are constant, and that the whole procedure of updating rating curves, as commonly done, is flawed and useless?

132: to further delimit HRUs. Which HRUs? Even reading the paragraph further, it is unclear how many HRUs are used. You say 4, but then mention "are mainly cropland and forest, whereas further south the land use is dominated by grassland", which are not in the 4 HRUs.

Section 3.3. This section, which is key to explain what was done in the paper, is very convoluted, and impossible to understand. The first sentence states "Parameters and process constraints have been applied to eliminate unrealistic model results". Which model results? "For example, the maximum storage" – why for example? I want to know exactly what was done and how it was done. Instead, there are just a few sentences of how the methodology was carried out, relegating the essential details to even more unclear supplementary materials. "The model was calibrated and evaluated" how was this done? What is the difference between calibration and evaluation? "For the evaluation of this calibration", why this? Is there another calibration? In general every paragraph contains a lot of information in a very convoluted way. It is necessary to describe the methodology in a much more streamlined way.

205. Needs to be expanded and clarified. • The procedure for calibration using h and the procedure for evaluation using Q needs to be clearly distinguished. • You write that you use d for model calibration and flow duration curves for model evaluation. Flow means Q, but all the plots show d duration curves. Where is the flow used? • The value dmod is not present in the Strickler formula (there is A and R). What is the relation to d? It should be written explicitly. • What is the relation between the Strickler formula and Q = a ∗ (h − h0)b? • What is the value of b? • If I understand well, the observed and modelled water discharge are obtained using the same formula with the same parameters. Why is then Qrec needed? Trying to explain 3 essential things (model calibration, evaluation and evaluation of rating curves) in the same paragraph does not work.

215. Does the model provide simultaneously the output at the 3 stations? Was it calibrated simultaneously to the 3 gauging stations? Or was it calibrated individually to each station? If it was calibrated individually to each station, shouldn't the parameters of the same HRU in different catchment be the same? How was this ensured?

230. It appears that the model was calibrated using FDCs. But the objective is to simulate streamflow. Are FDCs sufficient to represent streamflow time series? E.g. I can imagine that information about seasonality as well as timing of peaks is lost when calibrating to FDCs. How were these problems addressed?

230. I don't think you need caption for equations. The explanation should be in the main text.

230. Was it multi objective calibration leading to a Pareto-front? Needs to be clarified.

As the methodology was very unclear, it is difficult to understand and judge the value of the results. I guess major clarifications are needed, before a fair assessment can be made

---

## Referee Comment (RC2) · Anonymous Referee #2 · 22 Dec 2017

This paper applies FLEX-Topo to the Mara River Basin and calibrates the model using stage data. Developing methodologies to deal with discharge uncertainties, particularly in poorly gauged catchments, is an important research area and there is scope for the results presented to be of interest to the research community. However, at present, I struggle to understand what the key research contributions are from the paper and some of the methodology is difficult to follow. These points are expanded upon below.

Main Comments

1. The main aims and goals of the paper are poorly stated. Developing a hydrological model for a particular region as stated as the main goal of the paper is not a 'Cutting edge case study'. Similarly, the key research contributions from the paper are not clearly highlighted within the conclusions. The authors need to think about the novel

aspects of the paper and two-three key messages they want the reader to take away.

2. The title of the paper is currently misleading – I would remove 'with data uncertainty' as you do not consider uncertainties in stage data and the analysis of precipitation uncertainties is limited.

3. There needs to be a broader introduction to data uncertainty in the introduction including rainfall uncertainty as this is considered later in the paper. Furthermore, there should also be a larger section devoted to model calibration and model diagnostics and particularly how to perform robust model evaluation in the face of data uncertainties.

4. A separate section on data would be useful. At the moment, different datasets are introduced at lots of different points throughout section 2 and section 3.

5. One of the reasons for calibrating the model to water level is to 'avoid' uncertainties in water discharge. However, by then calibrating the 'c' parameter for the Strickler formula surely you just replace one source of uncertainty with another. As stated in the paper, it is likely that this parameter is also compensating for large sources of uncertainty in your precipitation data so I wonder how robust the results are given all these different sources of uncertainty. This needs to be better discussed in the limitations.

6. Section 3.3 is really difficult to follow and certain model choices need to be better justified –

a. Why was NSE chosen for model evaluation? How appropriate is NSE for calibrating water levels?

b. Strickler formula on line 205 needs to be presented as a separate equation – what do 'k' and 'i' denote?

7. Results

a. Section 4.1. The authors state at a couple of points that 'the observed and modeled water depth were quite similar to each other'. How similar is similar!? It would be better

here to state NSE values as a quantitative measure of how similar they are.

b. Section 4.2. How many point discharge measurements were taken? While these can be useful in model calibration and evaluation – I don't think comparing a single point measurement to a whole month of modeled discharge was useful and the fact that the modeled results were 'within an order of magnitude of the point measurement' not a particularly persuasive argument that the model was performing well. I think these could be incorporated much better into the model evaluation framework.

8. I was surprised that given the amount of effort that went into defining HRUs and different model structures for the basin based on field observations and interviews, no results or analysis was presented on these different model structures. Was it just data uncertainty that lead to poor model performance or also the definition of model processes? How were model simulations improved by using two different model structures rather than one?

---

## Author Comment (AC2) · 9 Feb 2018

The comment was uploaded in the form of a supplement:
https://www.hydrol-earth-syst-sci-discuss.net/hess-2017-661/hess-2017-661-AC2-
supplement.pdf

---

## Author Response (AR1)

We would like to thank the editor and all referees for their helpful comments. With these comments, we were able to improve the paper significantly. One of the major changes is that the main goal and key messages were more clearly focused. For this purpose, and as advised by the editor, the areal rainfall analysis was excluded throughout the paper. In addition, the method description were restructured and explained more detailed. These improvements have changed the manuscript significantly and therefor it is quite hard to indicate all individual changes. However, besides detailed answer to all comments of editor and referees (starting from page 2), we summarize the main changes we made below (linked to referees comments).

Objective, key messages and title

All reviewers and the editor indicated that the paper would benefit from clearer objectives and a more appropriate title. As a result, the main goal, key objectives and title were adjusted to:

- o Title: Rainfall-discharge modelling using river stage time series in the absence of reliable discharge information: a case study in the semi-arid Mara River Basin.
- o Main goal: The goal of this study is to illustrate the potential of water level time series for model calibration by incorporating the hydraulic equation describing the rating curve within the model.
- o Key objectives: 1) present an important data set for the Mara River Basin, 2) illustrate a hydrological modelling methodology where the model is calibrated using river water levels instead of discharge.

As a result of reformulating the key objectives, the analyses on the areal rainfall estimates was excluded as suggested by the editor to give the paper a clearer focus. Changes in the main objective and key messages were applied throughout the entire paper, for example in the introduction by reformulating the goal (page 2, lines 57-62), methods (rainfall analysis removed), results and discussion (rainfall analysis removed), and conclusion by reformulating the goal and key messages (page 17, lines 352-362).

In response to:

- o Editor: "*As all the referees highlight, and as you already acknowledge in your replies, the main objective should be made clear and the presentation should be structured along such line.*"
- o Referee #1: "*Section 1. The paper states that the "The goal is to develop a reliable hydrological model for the semi-arid and poorly gauged Mara". In my opinion, this is not the kind of objectives that warrants a publication. I am convinced that the authors can identify a set of more appealing objectives for their work.*"
- o Referee #2: "*1. The main aims and goals of the paper are poorly stated. Developing a hydrological model for a particular region as stated as the main goal of the paper is not a 'Cutting edge case study'. Similarly, the key research contributions from the paper are not clearly highlighted within the conclusions. The authors need to think about the novel aspects of the paper and two-three key messages they want the reader to take away. 2. The title of the paper is currently misleading – I would remove 'with data uncertainty' as you do not consider uncertainties in stage data and the analysis of precipitation uncertainties is limited.*"

Methods

We agree with the editor and reviewers that the methods section needed to be explained more detailed and restructured. For example, the sensitivity analysis for the HAND and slope thresholds was added (Section 3.1, page 6, lines 136-141); the procedure on applying process and parameter constraints was described more detailed and formulas were added (Section 3.3, page 10, lines 199-202); the model calibration approach was explained more detailed using a new flow chart (Figure 7) and restructured significantly (Section 3.4, page 12, line 214); and a sub-section on the rating curve analysis was added (Section 3.5, page 14, lines 242-252). In the sub-section on the model calibration, additional formulas and parameter values were included, as also an explanation on why the model was calibrated using FDCs (Section 3.5, page 12, lines 216-219). The sub-section on the rainfall analysis was excluded.

This is in response to:

- o Editor: "*The proposed approach for replacing the use of unreliable rating curves and how you use the Strickler equation is the core of the work and should definitely be explained much better. The description of the available data but especially of the calibration/validation procedure (very complex per se: three river sections, multiple objective functions, ..) also need to be thoroughly revised and integrated with a number of important clarifications also on the choices (necessarily subjective in some cases) that you made.*"
- o Referee #1: "*Section 3.3. This section, which is key to explain what was done in the paper, is very convoluted, and impossible to understand. [...] In general every paragraph contains a lot of information in a very convoluted way. It is necessary to describe the methodology in a much more streamlined way. [...] The procedure for calibration using h and the procedure for evaluation using Q needs to be clearly distinguished.*"
- o Referee K. Keshavarz: "*There is no information about the calibration process of either the FLEX-Topo model or the Manning-Strickler formula. What were the initial ranges of parameters? How many parameters have been calibrated? Did the authors used an optimization algorithm or an uncertainty-based method? What were the final ranges/values of parameters?*"

**Response to the Editor**

Thank you very much for your comments. These were taken into account to improve the paper.

Comments of the editor:
*I do believe that it's very important providing more and more information that may be used in regions where it is difficult to obtain reliable and long times series of meteo-hydrological data for calibrating rainfall-runoff models is crucial and proposing procedures tailored to improve the model implementation in such regions is definitely an important topic for our journal. As all the referees highlight, and as you already acknowledge in your replies, the main objective should be made clear and the presentation should be structured along such line.*
*I would suggest to remove the analysis on the areal rainfall estimates: I believe that focussing on the first one of the key-messages you list in your reply to Ref#1 and Ref#2 (and that automatically implies addressing also the second one) is already an ambitious objective and more than enough for the paper.*
*In fact I am afraid that adding also information and a cursory analysis on the important issue of the rainfall spatial field (that alone would need a separate detailed analysis, probably focussing in detail also on the features of the typical rainfall events that are expected in this specific part of the world if you want to add something to the vast literature on the subject) would not help to improve the clarity of the work, and you already need to add a number of details and clarifications, especially on the proposed procedure and on the model implementation, as required by the referees.*
*The proposed approach for replacing the use of unreliable rating curves and how you use the Strickler equation is the core of the work and should definitely be explained much better. The description of the available data but especially of the calibration/validation procedure (very complex per se: three river sections, multiple objective functions, ..) also need to be thoroughly revised and integrated with a number of important clarifications also on the choices (necessarily subjective in some cases) that you made.*

Response to the comments:
We agree the paper would benefit from a clearer objectives and key messages. We also agree that removing the areal rainfall analysis would result in a clearer focus. Therefore, the main goal is reformulated as follows: *The goal of this study is to illustrate the potential of water level time series for model calibration by incorporating a hydraulic equation describing the rating curve within the model*. The key messages are 1) present an important data set for the Mara River Basin, and 2) illustrate a hydrological modelling methodology where the model is calibrated using river water levels instead of discharge. This was applied throughout the entire paper; hence, the rainfall analysis was removed accordingly. The importance of the rainfall was only mentioned briefly in the limitations and recommendations. In addition, the calibration procedure was explained more detailed and restructured taking into account all comments.

**Response to Anonymous Referee #1**

Thank you very much for your review. Your detailed comments will be taken into consideration to improve the paper.

Regarding the major comments:

*Section 1. The paper states that the "The goal is to develop a reliable hydrological model for the semi-arid and poorly gauged Mara". In my opinion, this is not the kind of objectives that warrants a publication. I am convinced that the authors can identify a set of more appealing objectives for their work.*

All three reviewers pointed out that the paper could benefit from clearer objectives and subsequently a more appropriate title. We agree with the reviewers that the paper needs improvement here. We have submitted our study as a "cutting edge case study". According to HESS "**Cutting-edge case studies** report on case studies that require (a) broadening the knowledge base in hydrology as well as (b) sharing the underlying data and models. These case studies should be cutting edge with respect to the quality and diversity of data provided the soundness of the models employed, and the importance of the study objective."

We present both 1) an important and high quality data set for the data-poor Mara River Basin after detailed analysis of the available rainfall, river stage and discharge measurements and 2) an innovation in rainfall-runoff modelling using river water level time series for model calibration in absence of reliable discharge data which is often encountered in African river basins. In our opinion the latter contributes to the knowledge base of hydrology, in particular rainfall-runoff modelling. In addition, we analysed the influence of rainfall data averaging in semi-arid basins where the rainfall typically has a high spatial and temporal variability.

The main goal was not to merely develop a hydrological model, but to develop a modelling methodology which can help increasing the hydrological understanding in this poorly gauged semi-arid region using water level time series for calibration instead of discharge since the rating curve was of very poor quality. Hence, the challenge was to assess the water availability despite the poor data quality. In the Mara River Basin, there is limited data available, let alone a complete assessment of the data availability and quality. In addition, there are only limited hydrological models of this basin, therefore the understanding of the local hydrological processes is quite limited. Moreover, the absence of good quality discharge time series is not unique to this area, therefore assessing the possibility of calibrating on water levels instead of discharge is very useful for poorly gauged areas and should be explored more detailed in future studies. The advantage of water level time series is the higher availability as it is easier to measure and higher reliability since there is no calculation step in between (using a rating curve). In the future this could be combined with remotely sensed altimetry data.

In short, our key objectives are: 1) present an important data set for the Mara River Basin, 2) illustrate a hydrological modelling methodology where the model is calibrated using river water levels instead of discharge and 3) show the difference between input averaging of the rainfall as typically done and output averaging of the modelled discharge. The latter allows the inclusion of the non-linear behaviour of the rainfall-discharge relation in river basins.

Therefore the key messages for the reader to take away are:
1. In poorly gauged river basins, calibration on water level time series is more reliable than on discharge time series since additional uncertainties arise from fitting rating curves on scarce discharge measurements.
2. In this methodology, the water level-discharge relation is implicitly included in the model; the power exponent of this relation is related to the geometrical data which is observable in the field.
3. The method for dealing with highly spatially distributed rainfall in hydrological modelling is significant to obtain reliable results.

To take this comment into account and highlight these key objectives more clearly, this division into these main topics will be applied throughout the article. In combination with a clearer title, we hope the key messages will be clearer. We suggest changing the title into: **Rainfall-discharge modelling using river stage time series in the absence of reliable discharge information: a case study in the semi-arid Mara River Basin.**

*Section 1. Can the authors clarify why using water levels for model calibration avoids the effect of discharge uncertainties? This is presented as a fact, with no references to previous literature, and no explanations. I do not find the explanation obvious. Do they imply that rating curves are constant, and that the whole procedure of updating rating curves, as commonly done, is flawed and useless?*

Thank you for this comment, this indeed should be explained more explicitly.

It is important to make a distinction between well and poorly gauged river basin. In well gauged basins, sufficient discharge measurements can be available for fitting a rating curve more reliably and updating it regularly. In that case, discharge time series are indeed reliable and useful for model calibration. However, in poorly gauged areas, discharge measurements are generally very scarce. As a result, rating curves are fitted to scarce data and not updated regularly resulting in high uncertainties especially when extrapolating. As a result, there are significant uncertainties in discharge time series. Water level time series however are direct measurements which are therefore more reliable.

For this specific case of the Mara River, data analysis indicated that there are indeed high uncertainties in the discharge data (section 2). Therefore, here water level time series were more reliable than the discharge.

In short, using water levels for model calibration instead of discharge is only an improvement if the rating curve is indeed of poor quality, as often the case in poorly gauged areas.

*132: to further delimit HRUs. Which HRUs? Even reading the paragraph further, it is unclear how many HRUs are used. You say 4, but then mention "are mainly cropland and forest, whereas further south the land use is dominated by grassland", which are not in the 4 HRUs.*

The HRUs were defined in Line 134: "This resulted in four HRUs in the sub-basin of the Mara River Basin: forested hill slopes, shrubs on hill slopes, agriculture and grassland".

*Section 3.3. This section, which is key to explain what was done in the paper, is very convoluted, and impossible to understand. The first sentence states "Parameters and process constraints have been applied to eliminate unrealistic model results". Which model results? "For example, the maximum storage" – why for example? I want to know exactly what was done and how it was done. Instead, there are just a few sentences of how the methodology was carried out, relegating the essential details to even more unclear supplementary materials. "The model was calibrated and evaluated" how was this done? What is the difference between calibration and evaluation? "For the evaluation of this calibration", why this? Is there another calibration? In general every paragraph contains a lot of information in a very convoluted way. It is necessary to describe the methodology in a much more streamlined way.*

As the reviewer pointed out, the section on model methodology is indeed quite concise and could benefit from more elaboration. Therefore more details will be added in this section and in the supplements. A table of all the constraints was already included in the supplement (Table S1 and S2).

The formulation of "unrealistic model results" is indeed confusing, "unrealistic parameter sets" is more accurate as constraints were applied to eliminate unrealistic parameter sets rather than unrealistic model results; for instance forest interception should be greater than cropland interception. Furthermore, the model evaluation step consisted of several elements: first the model was evaluated by means of validation (which is what is meant in this section), later on by analysing the discharge on sub-catchment level, analysing the rating curves and the influence of the rainfall (in the discussion).

*205. Needs to be expanded and clarified.*
*1) The procedure for calibration using h and the procedure for evaluation using Q needs to be clearly distinguished.*
*2) You write that you use d for model calibration and flow duration curves for model evaluation. Flow means Q, but all the plots show d duration curves. Where is the flow used?*
*3) The value $d_{mod}$ is not present in the Strickler formula (there is A and R). What is the relation to d? It should be written explicitly.*
*4) What is the relation between the Strickler formula and $Q = a · L° U (h  h0)b$?*
*5) What is the value of b?*
*6) If I understand well, the observed and modelled water discharge are obtained using the same formula with the same parameters. Why is then Qrec needed? Trying to explain 3 essential things (model calibration, evaluation and evaluation of rating curves) in the same paragraph does not work.*

Reply to 1) There are indeed multiple steps in the use of water level and discharge for calibration and validation that could confuse the reader and should therefore be explained more clearly. First, the model was calibrated on water level (line 202), then the modelled discharge ($Q_{Strickler}$ and $Q_{mod}$) were compared to the recorded discharge (line 211) for model evaluation.

Reply to 2) The reviewer is right, this will be corrected. Instead of flow duration curve, the duration curves of the water depths were used for calibration. The flow was not used for the calibration.

Reply to 3) Thank you for this comment; this indeed is not written explicitly and should be included:

The cross-sections were simplified as a trapezium with a river width B and two different river bank slopes $i_1$ and $i_2$; these coefficients (Table 1) were estimated based on available cross-section data (Supplement S2). In addition, the water depth $d$ was calculated from the water level $h$ and reference level $h_0$.

$$A = B * d + \frac{1}{2} * d * (i_1 + i_2) * d$$

$$R = \frac{A}{B + d * \left((1 + i_1^2)^{\frac{1}{2}} + (1 + i_2^2)^{\frac{1}{2}}\right)}$$

$$d = h - h_0$$

**Table 1: Coefficients used for the simplification of the river cross-section**

|  | River width B [m] | River bank slope $i_1$ [-] | River bank slope $i_2$ [-] | Reference level $h_0$ [m] |
|---|---|---|---|---|
| **Amala** | 10.0 | 3.50 | 1.83 | 0 |
| **Nyangores** | 19.05 | 2.65 | 5.56 | 0 |
| **Mines** | 43.81 | 3.53 | 3.66 | 10 |

Reply to 4) Both equations estimate the discharge using water level data. In the rating curve (Q = a * (h-h0)^b), parameter *a* includes information on the cross-section, roughness and slope; parameter *b* information on the cross-section. This information is more direct in the Strickler formula.

Reply to 5) The value *b* varies for each cross-section. In line 206/7, this information could be included such as: Note that by using the Strickler formula the exponent of the rating curve is fixed; $Q = a * (h - h_0)^b$, with b = 1.71 at Amala, b = 1.71 at Nyangores and b = 1.70 at Mines using the same water level time series as for the calibration and validation.

Reply to 6) Thank you for this comment, this indeed needs to be explained more clearly. In contrast to what the reviewer stated, the modelled and observed discharge were obtained using different formulas. The modelled discharge $Q_{mod}$ was obtained through the FLEX-Topo model. $Q_{Str}$ was calculated using the Strcikler formula, a calibrated roughness/slope parameter *c* and water level time series. $Q_{rec}$ was obtained from the water department and was calculated locally probably by using a rating curve and the water level time series. This discharge $Q_{Str}$ was compared to $Q_{rec}$ to compare the recorded and modelled rating curves with each other.

*215. Does the model provide simultaneously the output at the 3 stations? Was it calibrated simultaneously to the 3 gauging stations? Or was it calibrated individually to each station? If it was calibrated individually to each station, shouldn't the parameters of the same HRU in different catchment be the same? How was this ensured?*
The reviewer has a good point here. The model was calibrated for all three stations individually using the same parameter ranges and constraints. As a result, the parameters were similar, yet slightly different for each station. After calibration, the "best" parameter sets were used for cross-validation. The model performed well when validating at Nyangores using the parameter set based on Mines ($NS_{FDC, log} = 0.94$ and $NS_{FDC} = 0.83$) whereas vice versa resulted in poor performance ($NS_{FDC, log} = 0.29$ and $NS_{FDC} = 0.00$). This is not surprising as all HRUs were represented when calibrating at Mines and only two HRUs when calibrating at Nyangores, namely forest and agriculture.

*230. It appears that the model was calibrated using FDCs. But the objective is to simulate streamflow. Are FDCs sufficient to represent streamflow time series? E.g. I can imagine that information about seasonality as well as timing of peaks is lost when calibrating to FDCs. How were these problems addressed?*
This is a good question. By calibrating on FDCs, the focus is on the flow statistics (e.g. how often high flows occur). This information is also in the streamflow, only the exact timings are not included when calibrating on FDCs. However, in this case, the timings were off anyway due to the limited number of rainfall stations available which was insufficient to capture the spatial heterogeneity well. Therefore, in this case the FDCs were good for model calibration.

*230. Was it multi objective calibration leading to a Pareto-front? Needs to be clarified.*
Thank you for this comment. Instead of analysing a Pareto-front, the values for the objective functions were ordered and the ones with the highest values were considered as "good" parameter sets.

**Response to Anonymous Referee #2**

Thank you very much for your review. Your detailed comments will be taken into consideration to improve the paper.

Regarding the major comments:
*1. The main aims and goals of the paper are poorly stated. Developing a hydrological model for a particular region as stated as the main goal of the paper is not a 'Cutting edge case study'. Similarly, the key research contributions from the paper are not clearly highlighted within the conclusions. The authors need to think about the novel aspects of the paper and two-three key messages they want the reader to take away.*
*2. The title of the paper is currently misleading – I would remove 'with data uncertainty' as you do not consider uncertainties in stage data and the analysis of precipitation uncertainties is limited.*
Reply to 1-2): All three reviewers pointed out that the paper could benefit from clearer objectives and subsequently a more appropriate title. We agree with the reviewers that the paper needs improvement here. We have submitted our study as a "cutting edge case study". According to HESS "**Cutting-edge case studies** report on case studies that require (a) broadening the knowledge base in hydrology as well as (b) sharing the underlying data and models. These case studies should be cutting edge with respect to the quality and diversity of data provided the soundness of the models employed, and the importance of the study objective."

We present both 1) an important and high quality data set for the data-poor Mara River Basin after detailed analysis of the available rainfall, river stage and discharge measurements and 2) an innovation in rainfall-runoff modelling using river water level time series for model calibration in absence of reliable discharge data which is often encountered in African river basins. In our opinion the latter contributes to the knowledge base of hydrology, in particular rainfall-runoff modelling. In addition, we analysed the influence of rainfall data averaging in semi-arid basins where the rainfall typically has a high spatial and temporal variability.

The main goal was not to merely develop a hydrological model, but to develop a modelling methodology which can help increasing the hydrological understanding in this poorly gauged semi-arid region using water level time series for calibration instead of discharge since the rating curve was of very poor quality. Hence, the challenge was to assess the water availability despite the poor data quality. In the Mara River Basin, there is limited data available, let alone a complete assessment of the data availability and quality. In addition, there are only limited hydrological models of this basin, therefore the understanding of the local hydrological processes is quite limited. Moreover, the absence of good quality discharge time series is not unique to this area, therefore assessing the possibility of calibrating on water levels instead of discharge is very useful for poorly gauged areas and should be explored more detailed in future studies. The advantage of water level time series is the higher availability as it is easier to measure and higher reliability since there is no calculation step in between (using a rating curve). In the future this could be combined with remotely sensed altimetry data.

In short, our key objectives are: 1) present an important data set for the Mara River Basin, 2) illustrate a hydrological modelling methodology where the model is calibrated using river water levels instead of discharge and 3) show the difference between input averaging of the rainfall as typically done and output averaging of the modelled discharge. The latter allows the inclusion of the non-linear behaviour of the rainfall-discharge relation in river basins.

Therefore the key messages for the reader to take away are:
1. In poorly gauged river basins, calibration on water level time series is more reliable than on discharge time series since additional uncertainties arise from fitting rating curves on scarce discharge measurements.
2. In this methodology, the water level-discharge relation is implicitly included in the model; the power exponent of this relation is related to the geometrical data which is observable in the field.
3. The method for dealing with highly spatially distributed rainfall in hydrological modelling is significant to obtain reliable results.

To take this comment into account and highlight these key objectives more clearly, this division into these main topics will be applied throughout the article. In combination with a clearer title, we hope the key messages will be clearer. We suggest changing the title into: **Rainfall-discharge modelling using river stage time series in the absence of reliable discharge information: a case study in the semi-arid Mara River Basin**

*3. There needs to be a broader introduction to data uncertainty in the introduction including rainfall uncertainty as this is considered later in the paper. Furthermore, there should also be a larger section devoted to model*

*calibration and model diagnostics and particularly how to perform robust model evaluation in the face of data uncertainties.*

Thank you for this comment. One of the objectives is indeed on the uncertainty caused by the rainfall heterogeneity, more specific: the difference between averaging of the input precipitation in contrast to averaging the output modelled discharge. Therefore, the introduction should indeed also include rainfall variability. However, in this study, uncertainties in the data for the Mara River Basin were pointed out rather than performing a complete uncertainty analysis to assess the influence of data uncertainty on the modelling results. Therefore, we feel that a section on model uncertainty analysis in the introduction is outside the scope of this article.

*4. A separate section on data would be useful. At the moment, different datasets are introduced at lots of different points throughout section 2 and section 3.*

The reviewer makes a good point here. All data should be introduced in section 2. Those newly mentioned in section 3 (DEM, land cover map, NDVI and remotely sensed precipitation) should have been introduced in section 2 as well. This will be done by subdividing section 2 into multiple sub-sections: Section 2.1 Site description (lines 77-85), Section 2.2 Ground measurements (lines 87-112) and Section 2.3 Remotely sensed data. The latter will be added to introduce remote sensing data that are now newly mentioned in section 3:

Section 2.3 Remotely sensed data
Besides ground measurements, also remotely sensed data were used for the model development. The catchment classification was based on the topography and the land cover. For the topography, a digital elevation map (SRTM) with a resolution of 90 m and vertical accuracy of 16 m was used (U.S. Geological Survey, 2014). The land cover was based on Africover, a land cover database based on ground truth and satellite images (FAO, 1998). Moreover, NDVI maps were used to define parameter constraints.

New information mentioned in section 2.3 will then be excluded from section 3 to avoid repetition.

*5. One of the reasons for calibrating the model to water level is to 'avoid' uncertainties in water discharge. However, by then calibrating the 'c' parameter for the Strickler formula surely you just replace one source of uncertainty with another. As stated in the paper, it is likely that this parameter is also compensating for large sources of uncertainty in your precipitation data so I wonder how robust the results are given all these different sources of uncertainty. This needs to be better discussed in the limitations.*

This indeed is a limitation of this methodology. However, in contrast to the discharge uncertainties, this is a parameter uncertainty that could be quantified more accurately. This is a recommendation for future studies. Therefore, a new section will be added in the discussion to highlight more clearly the short comings of this methodology (e.g. compensation of the slope-roughness parameter $c$ for non-closure effects) and recommendations for future studies (e.g. quantification of uncertainties in the parameter $c$, methodologies to constrain or estimate parameter $c$, analysis of the potential of water level based model calibration in well gauged basins to assess the uncertainties more reliably, determination of suitable objective functions for calibrating on water levels instead of flow etc.).

*6. Section 3.3 is really difficult to follow and certain model choices need to be better justified –*
*a. Why was NSE chosen for model evaluation? How appropriate is NSE for calibrating water levels?*
Thank you for this comment. In this case, NSE was chosen for model calibration and validation. However, it was not analysed how appropriate this is for calibrating on water levels. Therefore this is a good recommendation for future studies!
*b. Strickler formula on line 205 needs to be presented as a separate equation – what do 'k' and 'i' denote?*
Thank you for this comment. This will be clarified as such (line 205): […], where R is the hydraulic radius, A the cross-sectional area, k the roughness and i the slope; […]

*7. Results*
*a. Section 4.1. The authors state at a couple of points that 'the observed and modelled water depth were quite similar to each other'. How similar is similar!? It would be better here to state NSE values as a quantitative measure of how similar they are.*
A quantitative measure is indeed useful here. This was done in Table 6.

*b. Section 4.2. How many point discharge measurements were taken? While these can be useful in model calibration and evaluation – I don't think comparing a single point measurement to a whole month of modeled discharge was useful and the fact that the modeled results were 'within an order of magnitude of the point*

*measurement' not a particularly persuasive argument that the model was performing well. I think these could be incorporated much better into the model evaluation framework.*

In total, five point measurements were taken at three locations (see section 2). As there are only a few measurements and a significant time difference between the measurements (2014) and the model (1970s/1980s), it is not possible to use these measurements for model evaluation other than comparing the order of magnitude. If there would have been more measurements, then more accurate comparison methodologies would have been possible.

*8. I was surprised that given the amount of effort that went into defining HRUs and different model structures for the basin based on field observations and interviews, no results or analysis was presented on these different model structures. Was it just data uncertainty that lead to poor model performance or also the definition of model processes? How were model simulations improved by using two different model structures rather than one?*

Thank you for this comment. Analyses on the effect of using different model structures (lumped vs. semi-distributed) were done in an early stage yet the results were indeed not included in the paper. The during the model development, a lumped model structure was compared with a semi-distributed model using two different model structures. This comparison showed applying multiple model structures significantly improved the model, especially during validation (Table 2).

**Table 2: Model comparison: semi-distributed vs lumped for calibration (1988-1991) and validation (1985-1987)**

|  | Calibration | | Validation | |
|---|---|---|---|---|
|  | $NS_{FDC,log}$ | $NS_{FDC}$ | $NS_{FDC,log}$ | $NS_{FDC}$ |
| **Semi-distributed** | 0.91 | 0.71 | 0.74 | 0.93 |
| **Lumped (SSF model structure)** | 0.87 | 0.42 | 0.00 | 0.41 |
| **Lumped (HOF model structure)** | 0.90 | 0.62 | 0.09 | 0.12 |

**Literature**
FAO: Africover, GLCN, 1998.
Digital Elevation Map: www.earthexplorer.usgs.gov, 2014.

**Response to K. Keshavarz**

Thank you very much for your review. Your detailed comments will be taken into consideration to improve the paper.

Regarding the major comments:

*"As far as I have understood, the authors have used water level observations (dobs) to calibrate the model. So, Manning-Strickler formula has been implemented in the model (line 60) to simulate both discharge, Qmod, and water levels, dmod. Moreover, the authors have produced discharges based on dobs using Manning-Strickler formula and named it QStrickler. Then they compared the recorded observed discharge, Qrec, with QStrickler and Qmod in Figure 12. How did the authors produce QStrickler? Have you had information about the cross-section details at three locations indicated in Figure 12? The research method explanation is hard to follow and understand."*

Thank you for this comment. The methodology indeed is explained quite concisely and could benefit form more elaboration. Nevertheless, the reviewer understood the methodology and answered the question correctly: the discharge $Q_{Strickler}$ was indeed calculated by using the Manning-Strickler equation, cross-section data and a calibrated parameter for the roughness and slope. This was explained briefly in lines 204 – 210.

*"There is no information about the calibration process of either the FLEX-Topo model or the Manning-Strickler formula. What were the initial ranges of parameters? How many parameters have been calibrated? Did the authors used an optimization algorithm or an uncertainty-based method? What were the final ranges/values of parameters? Have you tried any other objective function rather than Nash-Sutcliffe? Why have the authors used two validation periods for Mines (lines 221-222)?"*

This is a good point as this was indeed not mentioned in the paper and should be included. For the calibration, the MOSCEM-UA algorithm was applied (Vrugt et al., 2003). No other objective functions have been tested, however several signatures were tested such as the hydrograph, logarithm of the hydrograph and slope of the flow duration curve. As no major differences were found in this case, this was not tested more detailed. Two validation periods were used for Mines to use as much data as possible taking into account the limited data availability.

To address this issue more detailed in the paper, the Table 3 and Table 4 will be added in the supplement and the sentence in line 201 will be adjusted to:

After having set up the model and defined the constraints, the model was calibrated applying the MOSCEM-UA algorithm (Vrugt et al., 2003) and validated.

**Table 3: Parameter ranges and optimal parameter sets**

| Parameters | Parameter ranges | Unit | Optimal parameter set | | |
| --- | --- | --- | --- | --- | --- |
| | | | Nyangores | Amala | Mines |
| $I_{max, F}$ | 0.2 - 2.7 | mm | 1.26 | 0.60 | 2.34 |
| $I_{max, A}$ | 0.6 - 6.0 | mm | 1.10 | 0.60 | 1.51 |
| $I_{max, G}$ | 0.7 - 3.6 | mm | 0.78 | 0.64 | 1.56 |
| $I_{max, S}$ | 0.3 - 2.0 | mm | 1.24 | 0.62 | 1.71 |
| $\beta$ | 0.5 - 2.0 | - | 1.88 | 0.62 | 1.32 |
| $T_{lag}$ | 0.5 - 1.5 | D | 1.45 | 1.44 | 1.46 |
| $K_{f,H}$ | 1 - 28 | d | 27.24 | 3.01 | 6.01 |
| $K_{f,T}$ | 1 - 28 | d | 12.02 | 2.01 | 3.05 |
| F | 0 - 15 | mm/d | 0.42 | 12.77 | 1.71 |
| c | Mines : 0 - 2.6 Nyangores: 0.4 - 1.6 Amala : 3.2 - 4.1 | $m^{1/3}/s$ | 0.89 | 3.40 | 1.31 |
| $S_{s,max}$ | 50 - 150 | mm | 99.87 | 106.04 | 141.98 |
| $S_{F/S}$ | 0 - 0.5 | - | 0.27 | 0.30 | 0.22 |
| $S_{A/G}$ | 0 - 0.5 | - | 0.09 | 0.33 | 0.24 |

**Table 4: Fixed parameters**

| Parameters | Parameter value | Unit |
|---|---|---|
| $K_s$ | 28 | d |
| $C_e$ | 0.5 | - |
| $S_{umaxF}$ | 122 | mm |
| $S_{umaxA}$ | 94 | mm |
| $S_{umaxG}$ | 83 | mm |
| $S_{umaxS}$ | 89 | mm |
| $S_{max,Amala}$ | 46 | mm |
| $S_{max,Nyangores}$ | 74 | mm |
| $S_{max,Middle}$ | 122 | mm |
| $S_{max,Lemek}$ | 119 | mm |
| $S_{max,Talek}$ | 69 | mm |
| $S_{max,Sand}$ | 29 | mm |
| $S_{max,Lower}$ | 48 | mm |

*"The time-step of the model seems to be neglected. The information about the timestep is not discussed in the paper expect a minor reference under Table 4 caption. Have you tried different time-steps? Could results improve if you use a smaller time-step?"*

The model was run on a daily time scale. A smaller time-step was not possible due to data limitations. As this was indeed not mentioned clearly in the paper, the sentence in line 201 will be adjusted to:

After having set up the model and defined the constraints, the model was calibrated on a daily time scale applying the MOSCEM-UA algorithm (Vrugt et al., 2003).

*"One of the main purposes of hydrological models is producing the hydrographs at different locations. Although authors have tried to indicate the water level time series (Figures 7, 8, 9 and 14), the hydrographs are missing."*

Discharge time series were indeed not shown as the focus was on simulating the water depth instead of the discharge. For comparison sake, this will be included in the supplement.

*The details of sensitivity analysis to produce thresholds of different landscape slopes and HAND values are missing. Is the HAND model based on the research of Nobre et al. (2011)? Have you used any specific sensitivity analysis algorithm/approach?*

Thank you for this comment. The thresholds influence the area contribution of the different landscapes, for instance a higher slope threshold could result in less hillslope areas. In the sensitivity analysis, this influence of the thresholds on the change of the area contribution was analysed. It was found that these area contributions behave asymptotically to changes in the thresholds. Therefore thresholds were chosen where changes in area contributions become insignificant. This asymptotical behaviour was strongly visible for the slope threshold. As there were no wetlands (based on field observations), the HAND threshold was set to zero; this will be corrected in the paper.

*Are calibrated roughness values in accordance with the streambed material for Manning-Strickler formula?*

Yes, they are. Natural channels with short grass typically have a Strickler coefficient between 25 and 45 $m^{1/3}$/s. The calibrated Strickler parameter was within this range assuming a slope between $10^{-2}$ and $10^{-4}$ which is realistic as it is a flat area with multiple rapids.

*How did the authors specify the average flow velocity (line 165)? Would changing this parameter value impact the overall results? Does it change the hypothesis of using Manning-Strickler formula?*

Thank you for this comment. The average flow velocity was an assumption which agreed with the point measurements in the river. With this velocity, the maximum delay from the sub-catchment furthest away was 4 days. Changing this velocity would change the timing of the flow from a specific sub-catchment. However, this timing uncertainty was insignificant compared to timing uncertainties caused by the highly heterogeneous rainfall which was poorly represented with the available stations.

Regarding the minor comments:

Thank you for those comments, they will be taken into consideration. These comments included: correcting English language, adding a separate section introducing the different data sources used, checking the literature

referencing to avoid missing or faulty references, renaming "Strickler formula" to "Strickler-Manning formula" to make it more general, referring to specific tables and figures in the supplement (e.g. see Table S1) instead of the supplement in general ("see supplement") and adjusting some figures. For the figures, sub-figures can be indicated more clearly through numbering/letters, months written out instead of numbers, sub-catchment boundaries included in the legend, figure adjusted such that the number of stations are consistent with the text.

In addition, to respond more detailed to some of the minor comments:

*The title of research seems awkward. What does 'modeling [: : :] with data uncertainty' mean? Where did the uncertainty of streamflow, either water level or discharge, come into consideration?*

Also other reviewers have stated that the title needs to be improved. The following title is suggested: Rainfall-discharge modelling using river stage time series in the absence of reliable discharge information: a case study in the semi-arid Mara River Basin.

*Equation 1 indicating Nash-Sutcliffe formula is wrong (lines 230 to 233).*

Unfortunately, it is not clear to the authors what is wrong with this equation. The caption however will be removed.

**Equation 1: Formulas for the Nash-Sutcliff objective function. The indices mod and obs indicate modelled and observed values, respectively. In all cases, sorted data was used for the calculation of the objective function therefore the flow duration curve was calibrated.**

$$NS_{\log(d)} = 1 - \frac{\Sigma\big(\log(d_{mod,sorted}) - \log(d_{obs,sorted})\big)}{\Sigma\big(\log(d_{obs,sorted}) - \log(d_{obs,avg})\big)} \qquad NS_d = 1 - \frac{\Sigma\big(d_{mod,sorted} - d_{obs,sorted}\big)}{\Sigma\big(d_{obs,sorted} - d_{obs,avg}\big)}$$

*What is the time period of discharge data indicated in Figure 12?*

In this figure, the model calibration results were plotted (see caption), therefore the time periods used were the ones used for calibration (lines 219-229).

*Section 4.4 needs more discussion as no general suggestion to future research is made. Moreover, it is not apparent whether these strategies have improved the results of calibration.*

Thank you for this comment. Indeed, a section will be added to the discussion to include details on limitations of this methodology (e.g. compensation of the slope-roughness parameter $c$ for non-closure effects) and recommendations for future studies (e.g. quantification of uncertainties in the parameter $c$, methodologies to constrain or estimate parameter $c$, analysis of the potential of water level based model calibration in well gauged basins to assess the quality and uncertainties more reliably, determination of suitable objective functions for calibrating on water levels instead of flow etc.).

*The conclusions need to be considered again as many ideas have been repeated from the introduction/abstract part. It could have been more concise and explicit.*

Thank you for this comment. The conclusion will be reformulated such that it is more concise and that key messages are stated more clearly.

**Literature**

[revised manuscript text omitted]

---

## Author Response (AR2)

We would like to thank the editor and all referees for their helpful comment. Based on these comments, we made the following changes to improve the paper:

- Section 1 (Literature review): Two paragraphs were included to explain how previous studies used water level time series or remotely sensed river characteristics such as altimetry or river width for model calibration to put this study and the novelty more into context.
  In response to:
  Editor: "*I do agree that you should definitely add a more comprehensive a state-of-the-art review of the literature presenting approaches that use water levels for calibrating rainfall-runoff models, explaining what differentiate your work from them, in order to justify its publication.*"
  Referee #1: "*Perhaps the proposed paper still present elements of novelty compared to these studies. However, it is the authors' responsibility to illustrate in which terms their analyses complement or advance these previous studies. Is their method new? If not, how does it compare to the methods proposed by Jian et al, 2017? Is it better or worse?*"
- Section 3.4 (Methods, Model calibration): A short explanation including references was added on why the Nash-Sutcliffe function was used.
  In response to:
  Referee #2: "*Section 3.4, page 12. You provide good justification for your use of the water level duration curve as an evaluation metric but still do not provide good justification for why you use the Nash-Sutcliffe Coefficient.* "
- Section 4.5 (Limitations): A paragraph was added on the advantages and disadvantages of the proposed calibration method. This includes limitations as a result of the constant cross-section assumption and additional uncertainties due to adding the slope-roughness parameter of the Strickler-Manning formula as calibration parameter.
  In response to:
  Editor: "*A description of the main features (advantages and disadvantages) of the proposed approach in comparison to the previous literature, may also help in improving the section (4.5) on the Limitations of the work, putting it in the international literature context.*"
  Referee #2: "*Section 4.5, page 19. The limitations are a little brief - these points need expanding alongside some references.*"
  Referee #2: "*I am still not totally convinced by the methodology - in particular the assumption that the cross section does not vary over time and additional uncertainties derived from calibrating additional parameters in the Strickler-Manning formula.*"
- Section 5 (Conclusion) was merged with Section 6 (Recommendations)
  In response to:
  Editor: "*I would suggest merging limitations and recommendations in a single section, dealing with both the limits and way forward.*"
  Referee #2: "*Section 6, page 20. Recommendations are also very brief and don't really warrant their own section - I would move these into your conclusions.*"

[revised manuscript text omitted]

In the Previous studies focused on assessing the uncertainty of rating curves (Clarke, 1999; Di Baldassarre et al., 2009) and their effect on model predictions (Karamuz et al., 2016; Sellami et al., 2013; Thyer et al., 2011). However, in the absence of reliable rating curves, remotely sensed river characteristics related to the discharge data, such as river width and water level time seriescan provide reliable and valuable information on the flow dynamics (Seibert et al., 2016) and therefore could be a good alternative for model calibration and validation. For instance, previous studies derived the discharge from remotely sensed river width (Revilla-Romero et al., 2015; Sun et al., 2015; Yan et al., 2015) or river water levels measured with radar altimetry (Michailovsky et al., 2012; Pereira-Cardenal et al., 2011; Ričko et al., 2012; Schwatke et al., 2015; Sun et al., 2012; Tourian et al., 2017). Radar altimetry observations of river water levels were also used to calibrate or validate hydrological models 
[revised manuscript text omitted]

---

## Author Response (AR3)

We would like to thank the editor for the helpful comments. Based on these comments, we made the following changes to improve the paper:

- Section 1 (Literature review): Two paragraphs (page 2/3) were adjusted to include additional model types such as data-driven models.

  Editor: "*I believe that the state-of-the-art on the use of water levels instead of discharge (that we asked for) may be more inclusive, for example a number of data-driven technique has considered water levels as model output, as may be seen in many works and reviews on the use of neural networks in hydrology.*"

- Section 4.5 (Limitations)

  The second paragraph was changed such that the focus is directly on comparing the approach in this paper with previous work similar to it.

  Editor: "*I would also revise the new paragraph at page 20/21: there is no need to cite again (as already done, and with the same words, at p. 3-4) the remote-sensing papers, but you should directly focus on the differences from the previous works that are more similar to your approach (Jian and Seibert).*"

- Entire paper: Minor language edits were done throughout the entire paper.

  Editor: "*I leave to the Authors to re-verify all the paper (language included).*"

[revised manuscript text omitted]